# OceanBench: A Benchmark for Data-Driven Global Ocean Forecasting systems

**Anass El Aouni**[1]*    **Quentin Gaudel**[1]    **Juan Emmanuel Johnson**[2]

**Charly Regnier**[1]    **Julien Le Sommer**[3]    **Simon Van Gennip**[1]    **Ronan Fablet**[4]

**Marie Drevillon**[1]    **Yann Drillet**[1]    **Pierre-Yves Le Traon**[1]

[1]Mercator Ocean International, Toulouse, France
[2]International Methane Emissions Observatory, UNEP, Paris, France
[3]Université Grenoble Alpes, CNRS, Grenoble, France
[4] IMT Atlantique, Brest, France

## Abstract

Data-driven approaches, particularly those based on deep learning, are rapidly advancing Earth system modeling. However, their application to ocean forecasting remains limited despite the ocean's pivotal role in climate regulation and marine ecosystems. To address this gap, we present OceanBench, a benchmark designed to evaluate and accelerate global short-range (1-10 days) data-driven ocean forecasting. OceanBench is constructed from a curated dataset comprising first-guess trajectories, nowcasts, and atmospheric forcings from operational physical ocean models, typically unavailable in public datasets due to assimilation cycles. Matched observational data are also included, enabling realistic evaluation in an operational-like forecasting framework. The benchmark defines three complementary evaluation tracks: (i) Model-to-Reanalysis, where models are compared against the reanalysis dataset commonly used for training; (ii) Model-to-Analysis, assessing generalization to a higher-resolution physical analysis; and (iii) Model-to-Observations, Intercomparison and Validation (IV-TT) CLASS-4 evaluation against independent observational data. The first two tracks are further supported by process-oriented diagnostics to assess the dynamical consistency and physical plausibility of forecasts. OceanBench includes key ocean variables: sea surface height, temperature, salinity, and currents, along with standardized metrics grounded in physical oceanography. Baseline comparisons with operational systems and state-of-the-art deep learning models are provided. All data, code, and evaluation protocols are openly available at https://github.com/mercator-ocean/oceanbench, establishing Ocean-Bench as a foundation for reproducible and rigorous research in data-driven ocean forecasting.

## 1 Introduction

Global ocean forecasting is a cornerstone of Earth system prediction. Accurate forecasts of ocean dynamics underpin a broad range of societal and scientific needs, from climate monitoring and carbon budgeting to maritime safety, fisheries management, and disaster response. Like its atmospheric counterpart, ocean forecasting has historically been dominated by numerical models based on physics that solve the governing equations of fluid motion, thermodynamics, and biogeochemistry.

---

*Correspondence to: aelaouni@mercator-ocean.eu

39th Conference on Neural Information Processing Systems (NeurIPS 2025) Track on Datasets and Benchmarks.

These models have matured significantly over recent decades and now offer multiscale forecasts at increasingly high resolution. However, they remain computationally intensive, dependent on complex data assimilation workflows, and slow to iterate with typical development cycles that stretch over years. Moreover, despite steady gains, key physical processes such as submesoscale turbulence, deep convection, or eddy-mean flow interactions remain underresolved or poorly parameterized.

By contrast, data-driven ocean prediction remains in its infancy. This is due in part to the inherent complexity of ocean dynamics, the challenges of sparse and heterogeneous observations, and a historical lack of investment relative to atmospheric science. These factors have contributed to the slower adoption of machine learning (ML) methods in operational ocean forecasting, despite their recent success in other Earth system domains. Contributing to this gap is also the absence of a standardized benchmark which hinders progress by making it difficult to assess model skill, diagnose failure modes, or identify promising approaches.

To address this gap, we introduce OceanBench: a community benchmark for global short-range ocean forecasting (1-10 days). OceanBench is built on curated datasets from operational forecasting systems, including model first-guesses [2], nowcasts, and associated atmospheric forcings data which is typically omitted from public reanalyses. It also integrates matched observational datasets from both satellite and in-situ sources, enabling evaluation in a realistic real-time forecasting framework. The benchmark defines two complementary evaluation tracks: a model-to-observation track using standard skill scores and a model intercomparison track that supports traditional and process-oriented diagnostics. Finally, OceanBench includes baseline results from both operational systems and recent ML-based models, along with open-source tools for data access, training, and evaluation. By establishing a standardized and extensible framework, OceanBench aims to foster reproducibility, encourage collaboration, and accelerate progress in AI-enabled ocean forecasting.

## 2 Related work

In recent years, the success of deep learning in atmospheric sciences has spurred interest in data-driven Earth system prediction. Enabled by advances in model architecture and the availability of large-scale reanalysis datasets, several ML models have achieved competitive or superior performance compared to traditional numerical weather prediction (NWP) systems. Notable examples include GraphCast, Pangu-Weather, and FourCastNet, which demonstrate strong forecasting skill across various deterministic and probabilistic metrics, particularly for medium-range forecasts (1-10 days) [22, 3, 25]. Much of this progress has been driven by standardized benchmarks such as WeatherBench and its successor WeatherBench2, which provide high-quality datasets, reproducible pipelines, and unified evaluation protocols [26, 27]. These benchmarks have been instrumental in enabling rapid model iteration, fair comparison, and transparent reporting of progress.

In the ocean domain, machine learning applications have gained traction more slowly. The inherent complexity of ocean dynamics, including nonlinear interactions across a wide range of spatial and temporal scales, poses unique challenges for data-driven modeling. These challenges are compounded by limited observational coverage, especially below the surface, and by the diversity of forcing mechanisms such as wind stress, tides, and boundary currents that influence ocean behavior. Nevertheless, recent work has begun to demonstrate the viability of ML for global-scale ocean forecasting. Emerging global models like GLONET, XiHe, and WenHai [2, 32, 6], along with regional models [19, 14], have shown that it is possible to learn from historical analyses and reanalyses, or even directly from observations to generate skillful forecasts of key ocean state variables [16]. These models operate efficiently at inference time and are often focused on short-range forecasting, where satellite observations are more abundant and predictability is higher.

Despite recent momentum, progress in ocean ML forecasting has been limited by the absence of shared datasets and evaluation protocols. Framework "OceanBench: The SSH Edition" had a first iteration which attempted to standardize benchmarking for ocean observations and simulations but they only focused on a very small subset of variables, i.e., sea surface height and sea surface temperature, over a very small region, i.e., Gulf-stream [21]. OceanBench builds on the principles established by domain-specific efforts (such as "OceanBench: The SSH Edition") with the completeness and

---

[2]In operational oceanography and data assimilation, the first-guess (also known as the background) is the short-term forecast from a previous model run, typically used as the initial estimate of the ocean state before assimilating new observational data. It represents the model's best estimate of the current conditions based solely on past information and dynamical evolution. During assimilation, this first-guess is combined with newly available observations to produce an updated analysis, or nowcast, that better reflects the true state of the ocean.

robustness of broader benchmarks (such as WeatherBench2), adapting them to the ocean domain through curated datasets, standardized metrics, and baseline models. By doing so, it is a foundational resource for the growing ocean forecasting community, supporting reproducible research, consistent and fair comparison across both physical and machine learning models, and diagnostic evaluation tailored to the unique challenges of ocean dynamics. As more ML and hybrid models are developed, they can be integrated into the benchmark over time.

## 3 Core OceanBench Forcing and Evaluation Systems

This section introduces the forecasting systems and data sources that underpin the OceanBench benchmark: 1) the initialization, 2), the baselines, and 3) the reference datasets. OceanBench evaluates both physics-based models, which constitute the foundation of operational oceanography (introduced in Appendix A), and data-driven models that learn to predict ocean dynamics directly from historical data using machine learning models. These systems vary in spatial resolution, temporal coverage, and forecasting methodology and encompass both autoregressive and direct prediction approaches. All models are initialized from a common ocean state and are subjected to the same atmospheric forcing fields, ensuring fair and synchronized forecast launches. Model performance is evaluated using a consistent set of reference data, which includes both reanalysis products, providing the best available estimates of the full 3D ocean state, and operational analysis fields, which offer high-frequency, observation-constrained estimates used in real-time systems. In addition, observational data sets support resolution-agnostic comparisons through standardized evaluation protocols inspired by operational oceanography. This unified framework combines physics-based models, machine learning forecasts, and reanalysis and analysis references, which support rigorous, reproducible, and comparative evaluation of global ocean forecasting systems. A complete description of the candidate models for machine learning, as well as the observation data used for evaluation, is provided in Appendix B.

### 3.1 GLORYS12 Global Ocean Physics Reanalysis: Training and Reference

GLORYS12 is a global eddy-resolving ocean and sea ice reanalysis produced by the Copernicus Marine Environment Monitoring Service (CMEMS), spanning from 1993 to the present. It is based on the NEMO ocean model [24], configured at a horizontal resolution of $1/12°$ (approximately 8km) with 50 vertical levels. The vertical grid is refined near the surface, with 22 levels in the upper 100m to better resolve upper-ocean variability. Atmospheric forcing is provided by the ECMWF (European Centre for Medium-Range Weather Forecasts) reanalyses, using ERA-Interim for earlier years and transitioning to ERA5 for recent periods [8, 18]. The data assimilation methodology combines a reduced-order Kalman filter [28], which ingests altimeter sea level anomalies, satellite sea surface temperatures, sea ice concentrations, and in situ temperature and salinity profiles, with a three-dimensional variational (3D-VAR) scheme [23] to correct large-scale temperature and salinity biases. GLORYS12 outputs daily and monthly mean fields of temperature, salinity, currents, sea level, mixed layer depth, and sea ice parameters on a regular global grid at $1/12°$ resolution. Within OceanBench, it is used both to train AI-based forecasting models and to evaluate their performance. Its high resolution, physical consistency, and integration of observations make it a strong reference for benchmarking both data-driven and physics-based systems.

### 3.2 GLO12 Global Analysis and Forecast System: Nowcast, Baseline, and Reference

GLO12 is an operational global ocean forecasting system built on the same NEMO-LIM3 configuration as GLORYS12, employing a $1/12°$ (8 km) horizontal grid, 50 vertical levels, and enhanced surface resolution to resolve upper-ocean and mixed-layer dynamics. Bathymetry is constructed using a blend of ETOPO1 and GEBCO8 datasets for depths between 200-300 m. Unlike GLORYS12, which is a delayed-mode reanalysis, GLO12 operates in near-real time. It is driven by high-resolution $(1/10°)$ ECMWF IFS forecasts that provide momentum, heat, and freshwater fluxes at sub-daily frequency. Tidal forcing is incorporated using prescribed constituents, and river discharge accounts for input from over 100 major rivers and polar ice sheet meltwater [7]. The GLO12 system employs the SAM2 data assimilation framework [31], combining a 4D SEEK filter [5], Incremental Analysis Updates (IAU) [4], and 3D-Var bias correction [10]. Observations assimilated include ODYSSEA SST, OSI SAF sea ice, AVISO sea level anomalies, and CORIOLIS in situ profiles. Deep-ocean constraints (below 2000 m) rely on WOA2013 climatology with non-Gaussian error formulations, and mass conservation is enforced via a global sea surface height constraint.

GLO12 produces both near-real-time analyses and 10-day forecasts on a rolling basis. Each week, a nowcast (i.e., ocean analysis field approximately 1-8 days behind real time) is generated and used to initialize daily-updated forecasts. The most recent 11-15 days are reprocessed to ensure temporal consistency across the analysis and forecast products. The outputs include 3D fields of temperature, salinity, and currents, as well as 2D fields such as sea surface height, sea ice parameters, and mixed layer depth, with daily, 6-hourly, and hourly resolutions. Within the OceanBench framework, GLO12 plays a triple role:

**Nowcast (Initialization):** Weekly nowcast snapshots from GLO12 are used to initialize all candidate AI forecasting models. These represent the best short-term estimate of the ocean state, incorporating both model dynamics and assimilated observations. Specifically, one nowcast per week is selected (every Tuesday, when the most accurate and up-to-date nowcast fields become available following data assimilation and quality control) to ensure consistency in model initializations across the 2024 evaluation period, while capturing seasonal variability. These nowcasts are distributed through OceanBench, as they are not available via the standard Copernicus catalog.

**Forecast (Baseline):** The operational 10-day forecasts issued from each nowcast serve as the baseline prediction, representing the performance of a physics-based state-of-the-art model.

**Analysis (Reference):** The corresponding GLO12 analysis fields are used as the reference product for evaluating forecast skill, providing an observation-constrained, quality-assured benchmark.

This unified use of GLO12 ensures that candidate models are initialized from a common ocean state, compared against the same forecast baseline, and validated against a consistent reference, thereby supporting robust, reproducible, and fair benchmarking across different modeling approaches.

### 3.3 IFS Atmospheric Forcing Fields: Operational Forcings

To ensure a consistent and realistic forcing environment across all candidate models, OceanBench provides operational atmospheric forecast fields from the ECMWF Integrated Forecasting System (IFS). These fields represent the actual atmospheric forecasts used in the real-time execution of the GLO12 system, thereby replicating the conditions under which operational ocean models are forced in practice. The IFS products used here correspond to the high-resolution (HRES) configuration of the ECMWF deterministic forecast, produced daily at a spatial resolution of approximately $1/10°$. The forcings span the full set of ocean-relevant surface variables, including: 10-meter wind components (U10, V10), surface air temperature and humidity, surface pressure, downward shortwave and longwave radiation, precipitation and evaporation, zonal and meridional wind stress, surface heat flux and freshwater flux components.

In OceanBench, these 10-days IFS forecast fields are made available alongside the GLO12 nowcasts, corresponding to the same initialization time, every Tuesday the the whole period of 2024. This unified setup ensures that all data-driven models are not only initialized from the same oceanic state but are also forced with the exact same atmospheric conditions as those used in GLO12 forecasts. By standardizing both the initial state and the atmospheric forcing, this framework provides a fair and reproducible foundation for evaluating the performance of candidate models. Moreover, it offers a realistic setup aligned with current operational forecasting protocols, allowing future deployment scenarios to be mirrored during benchmark evaluation.

## 4   Evaluation set-up and metrics

To comprehensively assess the forecasting skill of candidate ocean models, we adopt a unified evaluation framework that is both temporally and spatially consistent. The evaluation protocol ensures that all models are assessed under the same initialization frequency and forecast horizon while respecting their native resolution to avoid unfair penalization of coarser models. Through the materialization of a harmonized challenger dataset, this setup establishes a fair and meaningful basis for comparison across a wide range of modeling approaches. Table 1 summarizes the datasets and variables used in the evaluation framework. This includes datasets used as references (i) for ML-based model trainings, (ii) as forecast inputs to generate OceanBench challenger datasets, and (iii) for the evaluations of the challenger datasets.

Building upon this foundation, we implement a multifaceted evaluation strategy (illustrated in Figure 1) that captures different aspects of model performance. This includes (i) observation-based intercomparison, (ii) reference-model benchmarking, and (iii) process-oriented diagnostics derived

| Short name / Variable | Description / Source | (Spatial) dim. / type | Units | Dataset / Product ID |
|---|---|---|---|---|
| *(A) Ocean* | | | | |
| **Ocean spatial dimensions** | | | | |
| `lat` | Latitude | 1D | ° | GLORYS / GLO12 / Challengers |
| `lon` | Longitude | 1D | ° | GLORYS / GLO12 / Challengers |
| `depth` | Depth | 1D | m | GLORYS / GLO12 / Challengers |
| **Oceanbench challenger temporal dimensions** | | | | |
| `lead_day_index` | Lead Day Index | 1D | int(1-10) | Challengers |
| `first_day_datetime` | Forecast First Day Datetime | 1D | datetime | Challengers |
| **Ocean state variables** | | | | |
| `zos` | Sea Surface Height | 2D | m | GLORYS / GLO12 / Challengers |
| `thetao` | Temperature | 3D | °C | GLORYS / GLO12 / Challengers |
| `so` | Salinity | 3D | PSU | GLORYS / GLO12 / Challengers |
| `uo` | Zonal Current | 3D | $\mathrm{m\,s^{-1}}$ | GLORYS / GLO12 / Challengers |
| `vo` | Meridional Current | 3D | $\mathrm{m\,s^{-1}}$ | GLORYS / GLO12 / Challengers |
| **Ocean derived variables** | | | | |
| | Zonal Geostrophic Current | 2D | $\mathrm{m\,s^{-1}}$ | |
| | Meridional Geostrophic Current | 2D | $\mathrm{m\,s^{-1}}$ | |
| | Mixed Layer Depth | 2D | m | |
| | Lagrangian trajectory | 2D | ° | |
| **Observations** | | | | |
| Surface Currents | Lagrangian drifter velocities (DSMOI) | In-situ | $\mathrm{m\,s^{-1}}$ | INSITU_GLO_PHY_UVASSIM_DISCRETE_NRT_013_054 |
| Temperature & Salinity Profiles | Argo profiling floats | In-situ, multi-depth | °C / PSU | INSITU_GLO_PHYBGCWAV_DISCRETE_MYNRT_013_030 |
| Sea Level Anomaly (SLA) | Merged satellite altimetry | Gridded (L3) | m | SEALEVEL_GLO_PHY_L3_NRT_008_044 |
| Sea Surface Temperature (SST) | FNMOC GODAE SFCOBS dataset | In-situ surface | °C | DSFNMOC |
| *(B) Atmosphere* | | | | |
| **Atmospheric forcing fields** | | | | |
| `sotemair` | 2 m Air Temperature | 2D | K | IFS |
| `sowinu10` | 10 m Zonal Wind Component | 2D | $\mathrm{m\,s^{-1}}$ | IFS |
| `sowinv10` | 10 m Meridional Wind Component | 2D | $\mathrm{m\,s^{-1}}$ | IFS |
| `sosudosw` | Downward Shortwave Radiation | 2D | $\mathrm{W\,m^{-2}}$ | IFS |
| `sosudolw` | Downward Longwave Radiation | 2D | $\mathrm{W\,m^{-2}}$ | IFS |
| `sowaprec` | Precipitation Rate | 2D | $\mathrm{kg\,m^{-2}\,s^{-1}}$ | IFS |
| `sod2m` | 2 m Dew Point Temperature | 2D | K | IFS |
| `somslpre` | Mean Sea Level Pressure | 2D | Pa | IFS |

Table 1: Summary of all datasets used in OceanBench, including model input variables (ocean and atmospheric fields) and observation datasets for CLASS-4 validation. All observation products correspond to near-real-time (NRT) data from 2024. Variables marked as 3D have multiple vertical levels; 2D variables are surface fields. An OceanBench challenger dataset is a multidimensional spatio-temporal datacube (five variables over 3 spatial dimensions and 2 temporal dimensions). Derived variables are computed as part of OceanBench process-oriented diagnostics. Lead day refers to the time elapsed between the model initialization and the target forecast time. Forecast first day datetimes are the 52 Wednesdays of year 2024.

from physically meaningful variables. Together, these components provide a holistic view of each model's ability to reproduce observed ocean dynamics, maintain internal physical consistency, and generalize beyond the training regime.

## 4.1   Evaluation window and forecast initializations

In this benchmark, we evaluate all forecasting models for the full year of 2024. This period was selected for several reasons. First, 2024 lies well beyond the training periods of most AI-based models, which were typically trained on data up to 2019. This temporal gap provides a rigorous test of the generalization capacity and robustness of the models to out-of-distribution conditions. Second, the year 2024 offers a sufficiently long evaluation window to compute reliable statistics for both pointwise and process-oriented metrics. While some phenomena, such as marine heatwaves or extreme transport events, may benefit from longer multi-year analyses, a full annual cycle remains a strong baseline for most oceanographic diagnostics. To ensure a fair comparison across models, we initialize each forecast using the same nowcast on every Tuesday in 2024. From each initialization date, all models perform a 10-day forecast, enabling a consistent temporal sampling framework. This weekly cadence strikes a balance between computational efficiency and temporal resolution for skill evaluation. This evaluation strategy allows us to assess forecast skill under realistic deployment scenarios, where models must extrapolate far beyond their training horizon. It also avoids contamination from data seen during training, ensuring that metrics reflect true predictive skill rather than overfitting to historical conditions.

## 4.2   Evaluation grid and domain

To ensure a fair and resolution-consistent evaluation across models of varying spatial granularity, each model is assessed at its native resolution. Rather than regridding all model outputs to a common

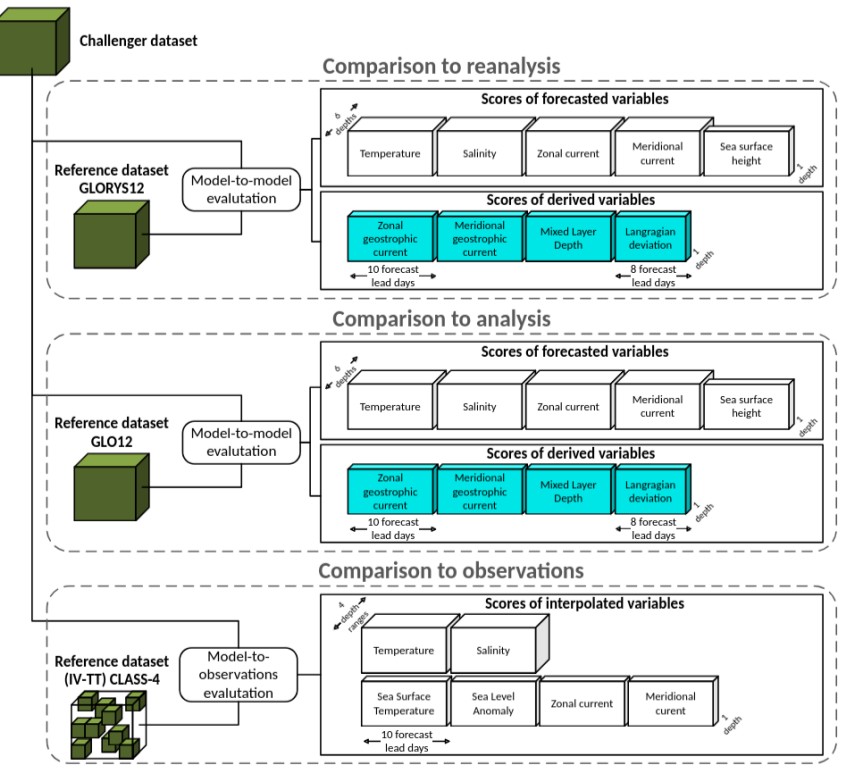

Figure 1: Overview of OceanBench's evaluation process.

target, we regrid the reference dataset, GLORYS12, to match the resolution of each challenger model. This approach avoids artificially inflating errors that could arise when comparing high-resolution truth with coarse predictions. By respecting each model's native resolution during evaluation, OceanBench avoids introducing biases and enables a clearer attribution of skill to the model architecture and training regime rather than its grid configuration. In contrast, CLASS-4 evaluation metrics [17, 29, 9] are inherently resolution-agnostic. They rely on comparisons against sparse observational data, such as along-track satellite or in-situ profiles, allowing for a direct and uniform comparison across models regardless of grid spacing. This dual evaluation strategy resolution-aware reanalysis comparison and resolution-invariant observation-based assessment provides a more balanced understanding of model performance across spatial scales, as it combines two complementary perspectives: one capturing the internal dynamical consistency of forecasts relative to a coherent reanalysis field, and the other assessing external realism against independent observations. Together, these perspectives help disentangle grid-dependent skill from true physical fidelity. All results reported in this manuscript are computed over the global ocean.

| Symbol | Range | Description |
|---|---|---|
| $f$ | | Forecast value |
| $o$ | | Observation value |
| $r$ | | Reference (GLORYS12 or GLO12) |
| $t$ | $1, \ldots, T$ | Evaluation time index |
| $l$ | $1, \ldots, L$ | Lead time index |
| $i$ | $1, \ldots, I$ | Latitude index |
| $j$ | $1, \ldots, J$ | Longitude index |
| $d$ | $1, \ldots, D$ | Depth index |
| $n$ | $1, \ldots, N$ | Lagrangian trajectory index |

Table 2: Evaluation metrics notations.

## 4.3 Models to observations evaluation track

The IV-TT CLASS-4 framework provides a benchmark for model validation by operating within the observation space, enabling a direct comparison between observed and modeled values across

both spatial and temporal dimensions. For each observation, the corresponding model counterpart is extracted at the same spatial and temporal location across various evaluation times and their corresponding forecast lead times, ranging from the best analysis (day 0) to ten-day forecasts. The CLASS-4 dataset includes observations of temperature and salinity from Argo profiles, sea surface temperature (SST) from surface drifting buoys, sea level anomaly (SLA) from along-track satellite measurements and surface current observations at 15m depth from GDP drifters buoys. This framework serves as a robust tool for intercomparison, facilitating a comprehensive assessment of the forecasting models' performance.

### 4.3.1 Root Mean Squared Error (RMSE)

RMSE is a standard metric to quantify the average magnitude of error between predicted and observed variables. It penalizes large errors more severely, making it particularly effective in identifying models that may occasionally produce large deviations from observations.

$$\text{RMSE}_{(l,d)} = \sqrt{\frac{1}{TIJ} \sum_t^T \sum_i^I \sum_j^J \left( f_{t,l,i,j,d} - x_{t,i,j,d} \right)^2}, \quad x \in \{o, r\} \tag{1}$$

### 4.4 Models to Analysis and Reanalysis tracks

This track performs a dense, global, and vertically resolved pointwise evaluation using the GLORYS12 and GLO12 ocean reanalysis and analysis as references. Unlike the models-to-observation track, which relies on sparse observational data, it leverages the high-resolution 3D structure of these products to assess full-field accuracy.

### 4.5 Process-Oriented Evaluation

Beyond conventional pointwise error metrics, process-oriented evaluation provides a deeper examination of the physical consistency and generalization ability of data-driven ocean forecasting models. Unlike traditional physics-based models that encode dynamical constraints through governing equations, neural network-based systems are primarily optimized to minimize loss functions, such as RMSE, often without explicit enforcement of physical laws. While this training approach can yield high pointwise accuracy, it does not guarantee that the resulting forecasts maintain coherent and physically plausible relationships between ocean variables.

To bridge this gap, we assess whether forecasted fields can be used to derive key oceanographic quantities that reflect underlying dynamics and processes. This includes diagnostics such as mixed layer depth (MLD) and geostrophic currents; none of which were directly used during training. These derived quantities provide a stringent test of physical realism, as they depend on accurate interplay between state variables like temperature, salinity, and sea surface height. Moreover, we evaluate trajectory coherence using Lagrangian diagnostics, which test whether velocity fields support realistic particle advection patterns over time.

This process-oriented approach thus serves as a critical complement to pointwise metrics, enabling a holistic assessment of whether a model not only forecasts individual variables accurately but also captures the physical integrity of the ocean system it aims to simulate. Details on the computation of these derived diagnostics including MLD, geostrophic currents, and Lagrangian trajectories are provided in Appendix C.

## 5 Benchmark Results

This section presents the headline results for the **Reanalysis Track**, which evaluates the short-range forecasting skill of each model against a high-quality reference reanalysis product. The summary scores, shown in Figure 2, offer a compact yet comprehensive view of model performance across key ocean state variables and diagnostic metrics. Results for the remaining benchmark tracks are provided in the Appendix D.

We assess five key physical state variables: potential temperature (thetao), salinity (so), and the zonal and meridional components of ocean velocity (uo, vo), all evaluated as three-dimensional fields at five standard depths (0.49 m, 50 m, 100 m, 200 m, 300 m, and 500 m). Sea surface height (zos) is evaluated alongside the surface-layer variables and included at the 0.49 m level to ensure

consistency in surface diagnostics. In addition to these pointwise state variables, process-oriented diagnostics are included to evaluate the physical realism and dynamical coherence of model outputs. These diagnostics comprise the mixed layer depth (MLD), geostrophic surface currents ($u_{geo}$, $v_{geo}$) derived from sea surface height gradients, and Lagrangian drifts deviation based on particle advection in surface velocity fields. The latter offers insight into the model's ability to preserve coherent flow structures and tracer transport over time. Collectively, the headline scores capture both state estimation accuracy and process-level fidelity, supporting a comprehensive evaluation of model performance across space, depth, and time.

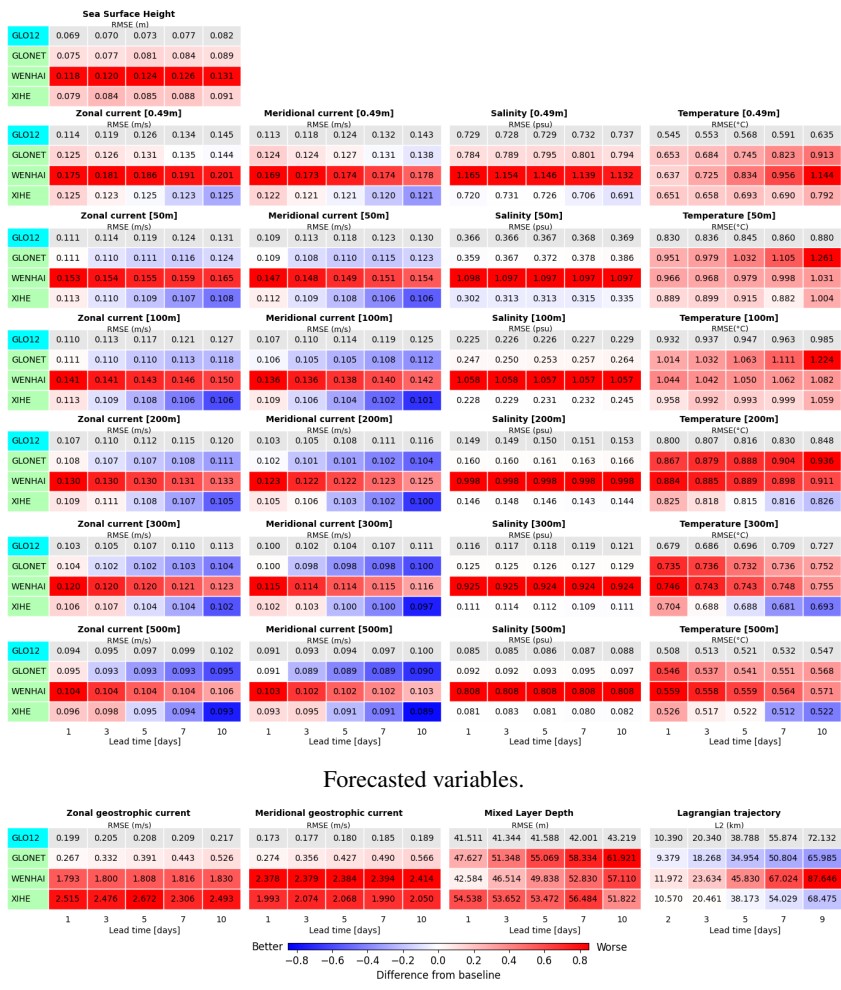

Figure 2: This table shows the absolute RMSE scores for the reanalysis track. These are the deterministic scores for the forecasted variables and physically-consistent diagnostic variables. Values show absolute RMSE. The colors denote % difference to the GLO12 baseline.

**ML versus Physics-based Models.** When comparing machine learning (ML) models to traditional physics-based systems, we observe that ML approaches tend to perform better on dynamical variables, particularly surface velocities (uo, vo). These variables reflect the advective structure of the flow field, which ML models appear to capture effectively from historical data. In contrast, ML models generally exhibit lower skill on scalar tracers such as potential temperature (thetao). However, their performance on salinity (so) is more variable and, in some cases, competitive or even slightly superior in specific tracks. This discrepancy may arisefrom the relatively low temporal variability and strong vertical stratification of tracers, which are more directly governed by conservation laws and vertical mixing processes; elements explicitly represented in physics-based models but only implicitly learned by ML models. Additionally, scalar tracers are more sensitive to atmospheric forcing in forecasting scenarios. Since such forcings are often more accurately represented in physical models

through prescribed boundary conditions, they may further contribute to the superior tracer predictions observed in physics-based systems.

**Performance Differences Among ML Models.** Among the ML-based approaches, we find significant performance variability, particularly in relation to the extent to which models incorporate explicit physical knowledge. Notably, WenHai, which includes bulk formulae and parameterizations similar to those used in operational systems, consistently ranks lowest among ML models across most metrics. This suggests that while physically inspired forcing mechanisms may add realism, they may also constrain the model's capacity to generalize or adapt to data-driven patterns when not fully integrated into the training pipeline. In contrast, more flexible data-driven architectures with less rigid physics priors achieve higher fidelity in reproducing reanalysis targets.

**Forecasting Strategy: Recursive versus Direct.** A key differentiator among ML models is their forecast design paradigm. XiHe, which employs a direct prediction strategy, forecasting future ocean states in a single forward pass, exhibits stable error evolution across lead times. In contrast, recursive models that predict iteratively (auto-regressively) tend to accumulate errors in a manner similar to traditional numerical forecast models. This compounding of small-scale inaccuracies over time limits their skill at extended lead times. These findings support prior observations in weather modeling that direct forecasting can mitigate temporal instability and better preserve long-range predictability.

**Impact of Physical Constraints.** Introducing physical constraints into machine learning (ML) architectures does not universally improve the accuracy of all forecasted variables. Among the ML-based systems, only the model that explicitly embeds physical constraints within its architectural design consistently outperforms others, most notably in the prediction of geostrophic surface currents ($u_{\text{geo}}, v_{\text{geo}}$). This suggests that architectural inductive biases aligned with physical principles, rather than external forcings or training objectives alone, are key to improving dynamical fidelity. For Lagrangian drift diagnostics, models tend to follow similar performance trends as those observed for surface velocity components ($u, v$), indicating that advection-dominated features are learned in tandem. However, the improved coherence in derived diagnostics such as $u_{\text{geo}}, v_{\text{geo}}$ further highlights the value of incorporating physically meaningful structure into ML model design.

## 5.1 Seasonal Dynamics of Forecast Skill

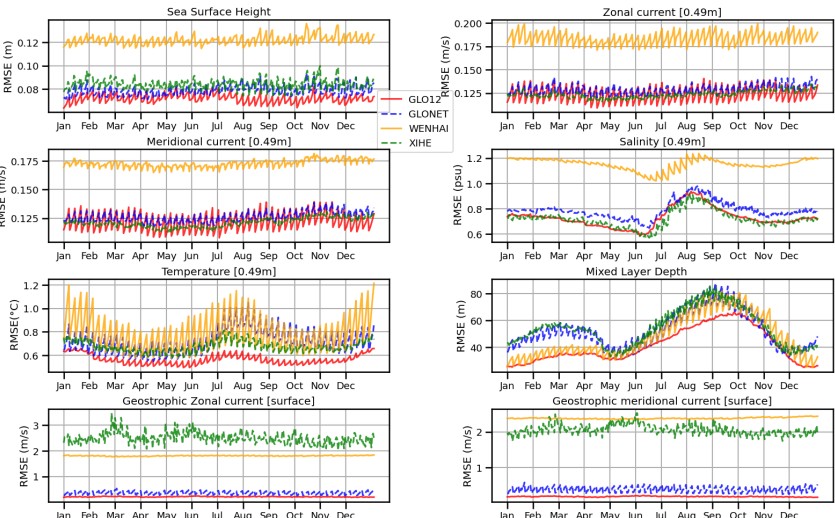

Figure 3: Time series of RMSE evolution throughout 2024 for all models in the Models-to-Reanalysis track.

To complement the headline skill scores, we further examine how model performance evolves throughout 2024. Specifically, we initialize a 7-day forecast every Tuesday and compute the corresponding RMSE for each forecast day. This produces a continuous time series of weekly forecast errors across the full year, providing insight into the temporal (and seasonal) modulation of model skill. We report, in Figure 3, the 7-day forecast RMSE computed daily throughout the year, stratified by variable type, providing a fine-grained view of each model's robustness over time.

**Dynamical Variables.** For core dynamical variables, zonal and meridional velocities $(u, v)$ and sea surface height (zos), we observe high-frequency fluctuations in RMSE that are consistent across all models and forecast cycles. These fluctuations likely reflect short-term variability and the accumulation of forecast error over the 7-day horizon. However, the absence of any systematic trend or recurring seasonal pattern suggests that model performance for these variables remains temporally stable and does not exhibit sensitivity to seasonal regimes. This holds for both physics-based and data-driven systems, indicating robustness of dynamical state prediction across seasonal transitions.

**Tracer Fields.** In contrast, strong seasonal patterns emerge in the forecast errors of tracer variables salinity (so) and temperature (thetao). All models, whether machine learning or physics-based, exhibit a clear seasonal modulation in RMSE. Salinity forecast errors tend to reach a minimum around mid-June and peak in September, while temperature forecast errors are lowest in mid-May and peak in August for most models. This modulation likely reflects seasonal changes in surface forcing, stratification, and vertical mixing. However, part of the observed variability, especially in salinity may also contain an interannual component, potentially linked to large-scale climate modes such as El Nino-Southern Oscillation (ENSO), particularly given the strong Nino conditions observed in 2024 [20]. Notably, the seasonal amplitude is less pronounced in the physics-based GLO12 system for temperature, possibly due to its regular data assimilation cycles and physical constraints. In contrast, ML-based systems show stronger seasonal variability, suggesting greater sensitivity to seasonal and possibly interannual variations in upper-ocean structure.

## 6   Conclusion

We introduce OceanBench, a unified benchmark for evaluating machine learning-based ocean forecasting models with a focus on scientific relevance and reproducibility. By leveraging consistent datasets, rigorous evaluation protocols, and metrics grounded in oceanographic practice, OceanBench aims to standardize model comparison across the community. Our initial release includes a suite of tasks that emphasize not only point-wise accuracy but also process-aware diagnostics, such as Lagrangian drift and mixed layer depth evaluation. OceanBench serves as a bridge between operational oceanography and modern data-driven methods, fostering cross-disciplinary collaboration.

**Limitation:** While OceanBench provides a robust and standardized framework for evaluating data-driven ocean forecasting models, several limitations remain in its current version. First, the benchmark focuses exclusively on deterministic forecasting tasks, with no support yet for probabilistic models or associated uncertainty-aware evaluation metrics, limiting its applicability for ensemble or stochastic forecasting approaches. Second, although OceanBench is designed for global-scale evaluation, it does not currently provide region-specific breakdowns of performance, which are critical for understanding model behavior in dynamically distinct areas such as western boundary currents or high-latitude regions. Third, the benchmark exclusively targets physical ocean variables such as sea surface height, temperature, and velocity without incorporating biogeochemical fields, which are increasingly important for monitoring marine ecosystems and carbon cycling. These limitations highlight opportunities for future extensions of OceanBench to support uncertainty quantification, regional diagnostics, and interdisciplinary ocean prediction tasks.

**Future Work:** We plan to extend the evaluation period to assess model performance over longer timescales and diverse oceanic conditions. New evaluation metrics will be incorporated to better capture spatially and dynamically varying forecast skill, giving appropriate emphasis to regions and depths of higher physical relevance. Additional measures will also address uncertainty quantification, extreme events, and physical consistency. Current OceanBench release is designed to accommodate different computational capacities, from GPU-limited research setups to large operational systems, and hence includes a coarser $1°$ resolution configuration enabling preparation for future updates that will further expand evaluation tracks to facilitate large-scale intercomparison and benchmarking. OceanBench is designed to remain a living benchmark, continually evolving to reflect the latest developments in ocean forecasting and machine learning. To support this vision, we maintain an open-source codebase, an up-to-date website, and an active GitHub repository. We encourage contributions and feedback from the broader community to help shape future versions of the benchmark and ensure its continued impact.

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

# A  Appendix: Preliminaries

This appendix provides key background information on ocean forecasting concepts, datasets, and terminology used throughout the paper. Operational oceanography is the provision of scientifically based information and forecasts about the state of the sea (including its chemical and biogeochemical components) on a routine basis and with sufficient speed, so that users can act on the information and make decisions before the relevant conditions have changed significantly or become unpredictable [15].

## A.1  Operation oceanographic systems

Operational oceanographic systems are historically based on numerical modelling of the ocean dynamic and data-assimilation schemes for the blending of the observations into the model to provide the most accurate description of the past and the future [30].

Figure 4 shows an overview of the lifecycle of an ocean state estimate at a given time t in the context of operational oceanography. In addition to observation assimilation, operational oceanographic systems are usually forced at the air-sea interface by atmospheric fields produced by operational atmospheric centers. Updated atmospheric fields and newly acquired observations are systematically released and thus used in the rerunning of the systems to produce more accurate ocean state estimates. As long as time t is in the future, the state estimate is called *forecast*. When time t is in the recent past and observations are assimilated in dribs and drabs, the state estimate is called *analysis*. After some point and depending on their operational constraints and data dissemination strategy, atmospheric and observation centers usually release best-quality datasets, allowing more updates of the state estimate, now called *reanalysis*.

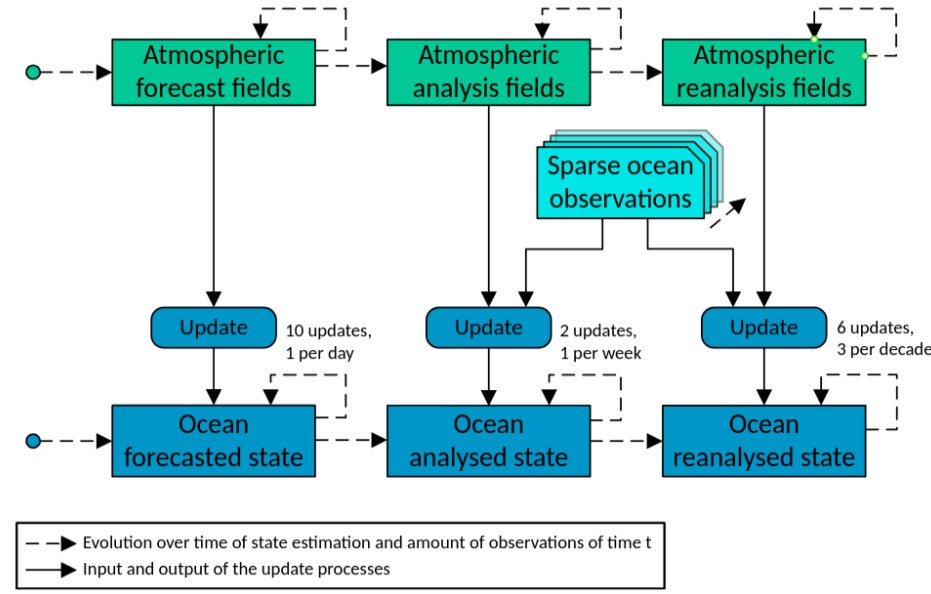

Figure 4: Lifecycle of an ocean state estimate at time t in operational oceanography: Operational oceanographic systems usually estimate ocean states by forcing atmospheric fields at air-sea interface and by assimilating observations acquired over time, enabling the production of more accurate state estimates. For future t (e.g., 10 days), the state estimate is a *forecast* and is updated regularly (e.g., daily) with new atmospheric forecasts. For recent past t (e.g., 2 weeks), the state estimate is an *analysis* and is updated regularly (e.g., weekly) with new atmospheric analysis and observations. Updates with high-quality atmospheric and observational datasets produce *reanalysis* (e.g., 6 updates over 2 decades). A system may include one or more of these components and the update frequencies may vary.

## A.2  Headline Metrics and Application Relevance

The ocean headline metrics exposed in Table 1 capture essential oceanic processes and are directly linked to a wide range of marine applications. The horizontal velocity components $(u_o, v_o)$ and sea

surface height ($z_{os}$) together describe the total surface circulation, encompassing both geostrophic and ageostrophic components that govern surface transport. These fields are critical for navigation, shipping route optimization, and the drift prediction of floating objects. Sea surface salinity ($s_{os}$) and temperature ($\theta_o$) represent thermohaline variability and air-sea interactions, underpinning applications in fisheries management, marine ecosystem monitoring, and climate diagnostics. The geostrophic current, derived from gradients in $z_{os}$, isolates the balanced circulation and mesoscale eddy activity, providing insight into nutrient transport and biologically productive regions relevant to fishing and aquaculture. The mixed layer depth (MLD) further characterizes vertical mixing and stratification, controlling heat and nutrient exchanges. Finally, Lagrangian trajectory assessments provide an integrated measure of transport skill, relevant for pollutant dispersion, search-and-rescue operations.

## B Appendix: ML Models and Observational Datasets

This appendix provides a detailed overview of the machine learning (ML) models benchmarked within the OceanBench framework, including their architectural designs, training protocols, and forecasting strategies. These models span a range of neural approaches tailored for global ocean prediction and are evaluated across the three benchmarking tracks defined in this study: models-to-analysis, models-to-reanalysis, and models-to-observation. In addition to describing the models, this appendix introduces the datasets used for the models-to-observations track, which assesses model forecast skill directly against independent, near-real-time ocean observations. This observation-based validation, grounded in the IV-TT CLASS-4 framework [17, 29, 9], provides an operationally relevant benchmark by leveraging high-quality drifters measurements.

### B.1 GLONET, A Neural Global Ocean Forecasting System

GLONET [2] is a data-driven, global ocean forecasting model targeting short-range (10-day) predictions of key ocean state variables, including 3D temperature and salinity, sea surface height (SSH), and surface currents. It operates at a horizontal resolution of $1/4°$ with 21 vertical levels, and is trained on GLORYS12 reanalyses interpolated to the target grid. The architecture follows a hybrid neural operator design that fuses multiple modeling paradigms. Large-scale patterns (e.g., gyres, equatorial currents) are captured using Fourier Neural Operators (FNOs), while CNNs enhance representation of finer-scale dynamics. A hierarchical transformer backbone models long-range spatial dependencies, particularly important for resolving land-sea boundaries and coastal complexities. The model follows an encoder-decoder structure that integrates multi-scale spatial and temporal features into a coherent latent representation. GLONET employs an autoregressive forecasting strategy over 10 days, where predictions are recursively used as inputs for subsequent steps. It does not perform online data assimilation; instead, it leverages the observational constraints already embedded in the GLORYS12 reanalyses used during training. For initialization, GLONET uses daily near-real-time analyses from GLO12. The system produces daily 10-day forecasts as daily mean fields, and outputs are experimentally distributed via the European Digital Twin Ocean (EDITO) platform. In OceanBench, GLONET represents the flagship AI-based model benchmarked against the physics-based GLO12 system, providing insight into the performance of deep learning approaches in operational forecasting contexts.

### B.2 XiHe: A Global Ocean Eddy-Resolving Forecasting System

XiHe [32] is a data-driven global ocean forecasting model designed to capture mesoscale and large-scale dynamics with high spatial resolution. It operates at $1/12°$ horizontal resolution with 23 vertical levels and is trained on 25 years of daily GLORYS12 reanalyses, enriched with near-surface wind fields from ERA5 and high-resolution SST from OSTIA. At its core, XiHe employs a hierarchical transformer architecture tailored for ocean forecasting. Custom ocean-specific self-attention blocks capture both local and global spatial dependencies, enabling the model to represent regional variability and inter-basin teleconnections. A land-ocean mask is applied to restrict learning to oceanic regions, improving spatial focus and reducing boundary artifacts. XiHe adopts a modular, temporally stratified design: 20 independent transformer-based models are trained separately for each forecast day (1 to 10) and vertical region (upper or lower ocean). This setup avoids the error accumulation common in autoregressive strategies and allows each model to specialize in lead-time and depth-specific dynamics.

## B.3 WenHai: Forecasting the Eddying Ocean with a Deep Neural Network

WenHai [6] is a global eddy-resolving ocean forecasting model based on deep learning, designed to predict upper ocean dynamics at $1/12°$ resolution across 23 vertical levels. Trained on 25 years of daily GLORYS12 reanalyses and ERA5 atmospheric forcings, WenHai focuses on mesoscale features such as eddies and sharp thermohaline gradients. Rather than predicting ocean state variables directly, WenHai forecasts their daily tendencies changes in temperature, salinity, sea surface height (SSH), and surface currents, which are applied recursively to update the ocean state over a 10-day forecast horizon. This tendency-based, autoregressive formulation emphasizes learning temporal dynamics. The model architecture is built on the Swin Transformer, leveraging localized self-attention to capture long-range spatial dependencies. Physical priors are embedded via bulk formulae for surface fluxes of momentum, heat, and freshwater. A volume-weighted loss prioritizes upper-ocean accuracy, aligning model training with regions of strong mesoscale variability and better observational coverage

| Model | Type | Autoregressive | Initialization | Horizontal Res. | Vertical Levels / Depth |
|-------|------|----------------|----------------|-----------------|------------------------|
| GLO12 | Physical (Forecast & Analysis) | Yes | Self-initialized (own forecast) + IFS | $1/12°$ | 50 levels (0m-seafloor) |
| GLONET | AI-based | Yes | From GLO12 | $1/4°$ | 21 levels (0m-seafloor) |
| XiHe | AI-based | No (Direct) | From GLO12 + IFS (u10, v10) | $1/12°$ | 23 levels (0-600 m) |
| WenHai | AI-based | Yes | From GLO12 + IFS (t2m, d2m, u10, v10, mtpr, ssr, strd, msl) | $1/12°$ | 23 levels (0-600 m) |

Table 3: Summary of the forecasting models used in OceanBench, including their type, initialization, and resolution. Autoregressive models produce forecasts iteratively from previous outputs, while direct models predict specific lead times independently.

Table 3 summarizes the key characteristics of the four forecasting systems considered in this study. Their computational requirements differ notably: the physics-based GLO12 model is the most demanding, as it integrates the full ocean dynamics at high spatial and temporal resolution using large-scale HPC resources. In contrast, the ML-based models require substantial resources during training but are considerably more efficient during inference, producing multi-day forecasts in a fraction of the time. This distinction highlights the potential advantages of data-driven approaches for scalable and real-time global ocean prediction.

## B.4 IV-TT CLASS-4 Observation Dataset

The CLASS-4 framework, developed by the Intercomparison and Validation Task Team (IV-TT), defines a standardized and operationally relevant protocol for assessing ocean forecasting systems within the observation space [17, 29, 9]. By directly comparing model outputs to near-real-time, independent observations at coincident spatial and temporal locations, this approach enables an unbiased evaluation of forecast skill across multiple variables and lead times, ranging from day 0 (best analysis) to 10-day forecasts.

Adopted in OceanBench, this framework complements analysis- and reanalysis-based evaluations by anchoring performance assessment in real observations, thereby supporting both scientific benchmarking and operational utility. The observational period considered spans the year 2024, with all datasets produced in near-real-time mode, thus aligning with the CLASS-4 philosophy of independence, timeliness, and applicability to operational oceanography.

The following observation datasets are employed for CLASS-4 validation:

- **Surface currents:** Validated against Lagrangian drifter velocities from INSITU_GLO_PHY_UVASSIM_DISCRETE_NRT_013_054 [DSMOI], which provides quality-controlled, near-real-time measurements from the global drifter array.

- **Temperature and salinity vertical profiles:** Sourced from the Argo program via INSITU_GLO_PHYBGCWAV_DISCRETE_MYNRT_013_030 [11], which offers multi-depth, multi-parameter observations from autonomous profiling floats.

- **Sea level anomalies (SLA):** Evaluated using gridded satellite altimetry from SEALEVEL_GLO_PHY_L3_NRT_008_044 [12], a Level 3 near-real-time product merging multiple satellite tracks.

- **Sea surface temperature (SST):** Assessed using in-situ measurements from the FNMOC GODAE SFCOBS dataset [ DSFNMOC], distributed via the GODAE Monterey Server and compiled from ships, moored and drifting buoys, and Coastal-Marine Automated Network (CMAN) stations.

# C Appendix: Derived Physical Diagnostics

This appendix outlines the methodology used to compute key derived quantities for process-oriented evaluation of ocean forecasts. These diagnostics, Mixed Layer Depth (MLD), geostrophic currents, and Lagrangian trajectories serve as physically meaningful benchmarks for assessing the internal consistency and dynamical realism of model outputs. While not directly optimized during training, these variables are inferred from predicted state fields (e.g., temperature, salinity, sea surface height, velocity) and thus provide a stringent test of whether neural forecasting systems capture the underlying physical processes of the ocean. The following subsections detail the mathematical formulations and computational procedures used to derive each diagnostic.

### C.0.1 Mixed Layer Depth (MLD)

MLD is a key indicator of ocean vertical mixing and stratification. Accurately predicting MLD is essential for simulating air-sea interactions, heat exchange, and biological productivity. MLD is derived from forecasted temperature and salinity profilesm and is commonly defined based on a density threshold criterion, such that the mixed layer is the depth at which the density difference from the surface equals a specified threshold. The MLD can be approximated as:

$$\text{MLD} = \min\left\{z \mid \rho_z - \rho_0 \geq \Delta\rho\right\} \tag{2}$$

where $\rho_z$ represents the density at depth $z$, $\rho_0$ is the density at the surface, and $\Delta\rho$ is a threshold value typically set to a small increment (e.g., $0.03 \ kg/m^3$) to capture the mixed layer's depth relative to surface conditions.

### C.0.2 Geostrophic Currents

Derived from sea surface height, geostrophic currents provide a diagnostic of large-scale ocean circulation. Accurate prediction of these currents is critical for understanding ocean transport and dynamics. Geostrophic currents are derived from forecasted SSH under the geostrophic approximation:

$$\mathbf{v}(\phi, \theta, t) = gf^{-1}\nabla^\perp\eta(\phi, \lambda, t) \tag{3}$$

where $g$ is the acceleration of gravity, $f$ presents the Coriolis coefficient, and $\eta(\phi, \lambda, t)$ is the sea surface height (SSH), which serves as a noncanonical Hamiltonian for surface velocity. $\perp$ stands for a 90° anticlockwise rotation of the gradient vector, producing a perpendicular flow direction as dictated by geostrophic balance.

### C.0.3 Lagrangian Trajectory

Lagrangian drift analysis offers insight into a model's ability to capture the advection of ocean particles over time, which is critical for applications involving transport processes such as pollutant dispersion, larval connectivity, and passive tracer dynamics. By simulating the motion of synthetic particles advected by model-predicted velocity fields, we assess whether the flow structures are coherent and physically realistic. Let's consider the ocean currents field:

$$\mathbf{v}(\mathbf{x}, t), \quad \mathbf{x} \in \mathbb{R}^2, \quad t \in [t_0, t_f] \tag{4}$$

and its associated ordinary differential equation:

$$\dot{\mathbf{x}} = \mathbf{v}(\mathbf{x}, t), \quad \mathbf{x} \in \mathbb{R}^2, \quad t \in [t_0, t_f] \tag{5}$$

where $\mathbf{v}$ the U and V components of ocean currents, defined on a possibly time-dependent spatial domain $\mathcal{U}(t) \in \mathbb{R}^2 \times [t_0, t_f]$.

Lagrangian trajectories are defined as:

$$\mathbf{x}(t_f, t_0, \mathbf{x}_0) = \mathbf{x}_0 + \int_{t_0}^{t_f} \mathbf{v}(\mathbf{x}(\tau), \tau)\, d\tau \tag{6}$$

To quantitatively evaluate the fidelity of Lagrangian trajectories, we compute the Euclidean distance between model-predicted and reference (GLORYS12) particle positions at each time step. It is

expressed in kilometers and averaged over all particles:

$$\text{Lagrangian drift deviation}(t) = \frac{1}{N} \sum_{n}^{N} \left| \mathbf{x}_i^f(t) - \mathbf{x}_i^r(t) \right| \tag{7}$$

This metric provides a time-resolved diagnostic of trajectory divergence, helping identify whether modeled flow fields maintain coherent transport pathways. Small Lagrangian errors suggest a physically plausible flow structure, which is particularly important for data-driven models not constrained by conservation laws.

## D   Appendix: Benchmark Track Results and Model Intercomparison

This appendix presents a consolidated analysis of model performance across the two core benchmarking tracks defined in OceanBench: models-to-analysis and observations-to-analysis. It brings together a comprehensive intercomparison of forecasting approaches, examining their behavior across spatial and temporal scales through a range of qualitative and quantitative diagnostics. The goal is to provide deeper insight into the strengths and limitations of each model in capturing ocean dynamics, fostering a more nuanced understanding of their generalization ability under realistic forecasting scenarios.

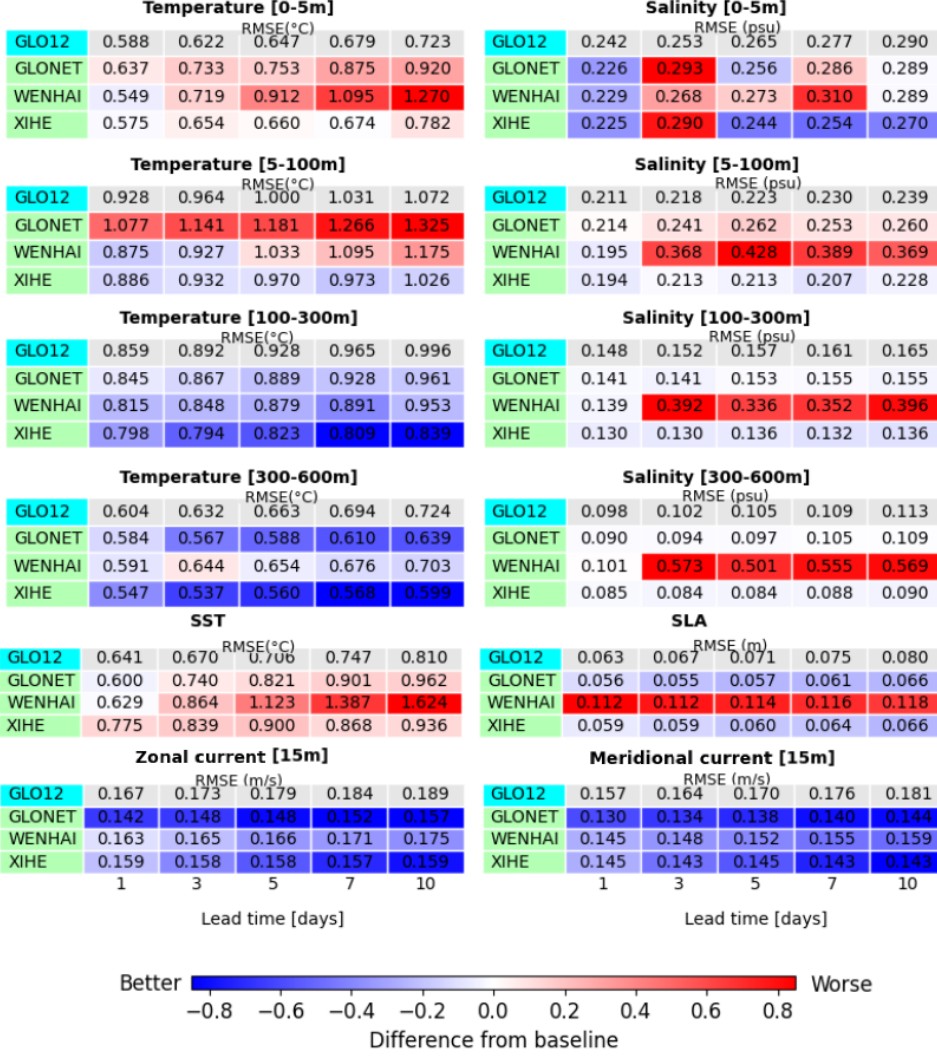

Figure 5: Models to observations track.

## D.1 Models-to-Observations Track

The models-to-observations track provides a direct evaluation of forecast skill against independent in situ and satellite observations (see Figure 5). Among the assessed variables, surface ocean currents, evaluated at a reference depth of 15 meters, exhibit notable skill differentials between modeling approaches. ML-based models demonstrate superior performance in this regime, with GLONET in particular achieving consistently lower errors relative to both traditional physics-based and other ML-based models. This improved performance likely stems from the capacity of ML models to capture advective structures and mesoscale variability present in historical training data.

The skill observed in surface velocity fields is mirrored to some extent in sea level anomaly (SLA) forecasts, which are used to derive geostrophic surface currents. Certain ML models also achieve competitive performance in SLA prediction, indicating an emerging ability to learn coherent surface dynamics from data alone, though the degree of success varies across architectures.

Forecast skill in temperature and salinity fields reveals distinct depth-dependent behavior. For temperature, ML performance tends to improve with depth, particularly at intermediate levels (e.g., 100-300 m), where thermocline structure is both stable and predictable. This suggests that data-driven models can internalize persistent stratification patterns when supported by sufficient historical context. In contrast, salinity forecasts tend to degrade with increasing depth, likely reflecting the more heterogeneous and patchy nature of salinity fields, which pose greater challenges for interpolation and learning. While regional breakdowns of performance are not available in the present analysis, it is reasonable to hypothesize that ML-based models gains are more pronounced in regions characterized by high mesoscale activity and dense observation coverage. Further spatial disaggregation would be required to confirm such patterns.

## D.2 Models-to-Analysis Track

The models-to-analysis track evaluates model forecasts against the GLO12 analysis (see Figure 6). Unsurprisingly, all models underperform relative to the GLO12 baseline in this track, as the reference analysis is itself generated from the GLO12 forecast model. Specifically, the GLO12 analysis is produced through a weekly data assimilation cycle applied to GLO12 forecasts, meaning it inherits both the dynamical structure and biases of the underlying numerical model. This setup creates a structural advantage for GLO12-consistent models, and correspondingly poses a higher bar for systems that diverge in design. This structural bias is clearly reflected in the results: WenHai, which incorporates physically inspired components such as bulk formulae for surface forcing, exhibits error evolution patterns that closely mirror those of GLO12, particularly for surface currents.

Such similarity suggests that shared physical assumptions lead to convergent dynamical behavior under this evaluation framework. In contrast, more flexible data-driven models tend to display different error trajectories, with some demonstrating improved accuracy at longer lead times, potentially due to a reduced coupling with the reference model's assimilation dynamics. For scalar variables such as temperature and salinity, however, no clear systematic trends emerge across models. This may reflect the more complex vertical structure and reduced observational constraint at depth, which weaken the influence of both physical priors and learned data patterns in shaping model skill under this benchmark.

In summary, the models-to-analysis track is best interpreted as a measure of structural consistency with the GLO12 system, rather than as an unbiased indicator of real-world forecast skill. It complements the observation-based track by revealing how different model classes align or diverge from an established operational baseline, and underscores the importance of using multiple benchmarks to robustly assess forecast performance across frameworks.

### D.2.1 Temporal Structure of Forecast Errors

To complement the overall skill metrics reported in the Models-to-Analysis Track, we analyze the temporal evolution of model performance over the full calendar year of 2024 as shown in Figure 7. This evaluation provides a time-resolved perspective on how forecast accuracy evolves in relation to both seasonal and sub-seasonal variability, using daily 7-day RMSE scores stratified by variable type. Compared to the Reanalysis Track, models appear more tightly clustered in performance, both at large scales (seasonal modulations) and at higher frequencies, where alignment with the weekly forecast cycle of the GLO12 analysis becomes apparent.

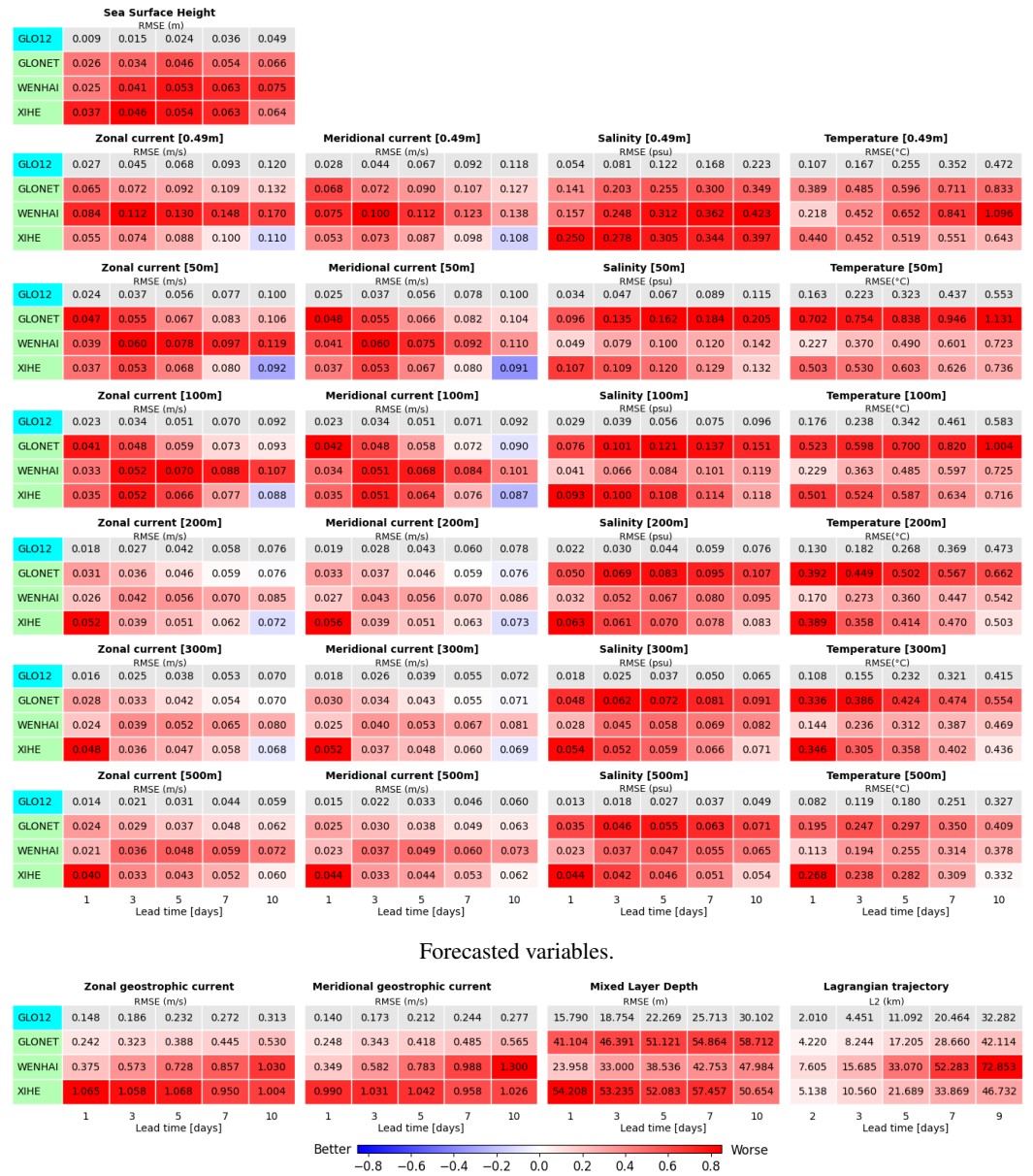

Forecasted variables.

Diagnostic variables.

Figure 6: Models to analysis track.

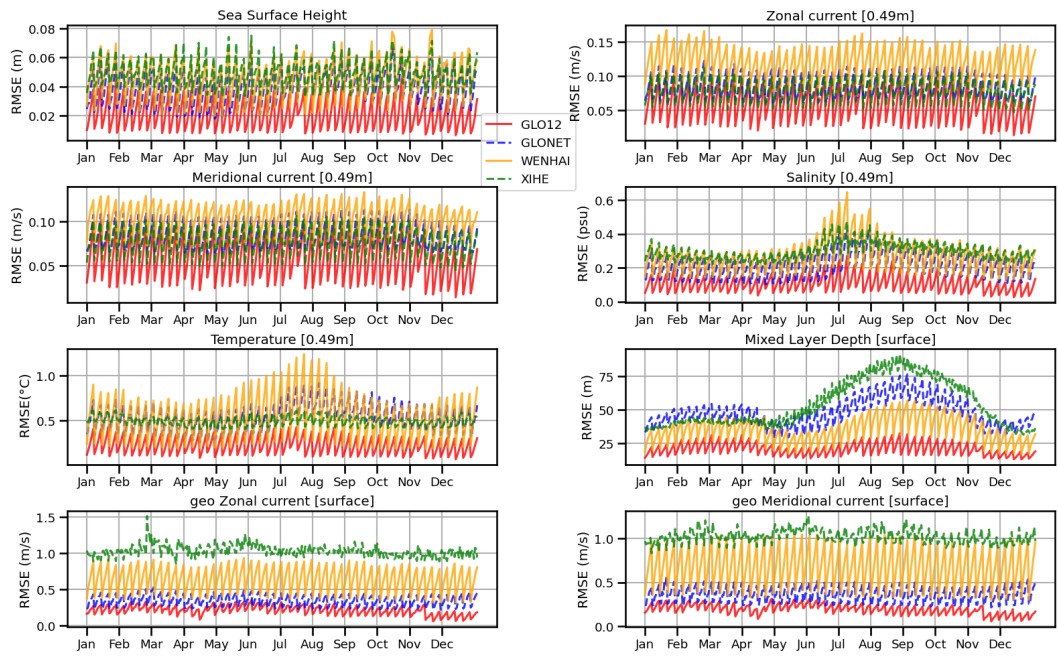

Figure 7: Time series of RMSE evolution throughout 2024 for all models in the Models-to-Analysis track.

**Geostrophic Currents and Dynamical Fields.** For geostrophic surface currents, the spread in error among models is markedly reduced compared to the Reanalysis Track. This convergence may reflect the structural imprint of the GLO12 forecast system on the analysis product, which effectively narrows the range of permissible dynamical behaviors. While short-term error fluctuations are still observed, likely tied to the 7-day assimilation cycle, the relative ranking of models remains consistent with the Reanalysis evaluation.

**Temperature and Salinity.** Seasonal modulation in tracer forecast errors is also diminished relative to the Reanalysis Track. Whereas clear annual cycles were previously observed, particularly strong in machine learning models, temperature and salinity RMSEs now exhibit weaker amplitude and reduced variability across models. Notably, the GLO12 baseline displays little to no seasonal pattern, likely due to its assimilation-driven correction toward climatological states. This damping effect appears to propagate into the analysis, thereby reducing the sensitivity of evaluation metrics to seasonal forcing signals. As a result, the Models-to-Analysis Track offers a more constrained and homogenized assessment of model fidelity, shaped in part by the characteristics of the reference itself.

## E   Spatial Structure and Scale-Resolved Evaluation

Beyond aggregate scores and temporal trends, spatial diagnostics provide essential insight into the qualitative behavior of ocean forecasting models. This section presents a series of spatially explicit analyses that complement the benchmark metrics by offering a visual and scale-aware assessment of model fidelity. These diagnostics not only reveal how errors manifest across different oceanographic regimes but also highlight the structural differences in model output, particularly in terms of resolved spatial scales and noise characteristics.

### E.1   Visual comparison of model outputs.

Qualitative analysis of the model outputs reveals a high degree of similarity in the spatial distribution and dynamical structure across all evaluated systems (see Figure 8). Core ocean state variables including sea level anomaly, surface temperature, salinity, and surface currents, exhibit coherent mesoscale features and basin-scale gradients that are well captured by all models, despite differences in resolution or architectural design. This convergence suggests a shared ability to reconstruct the

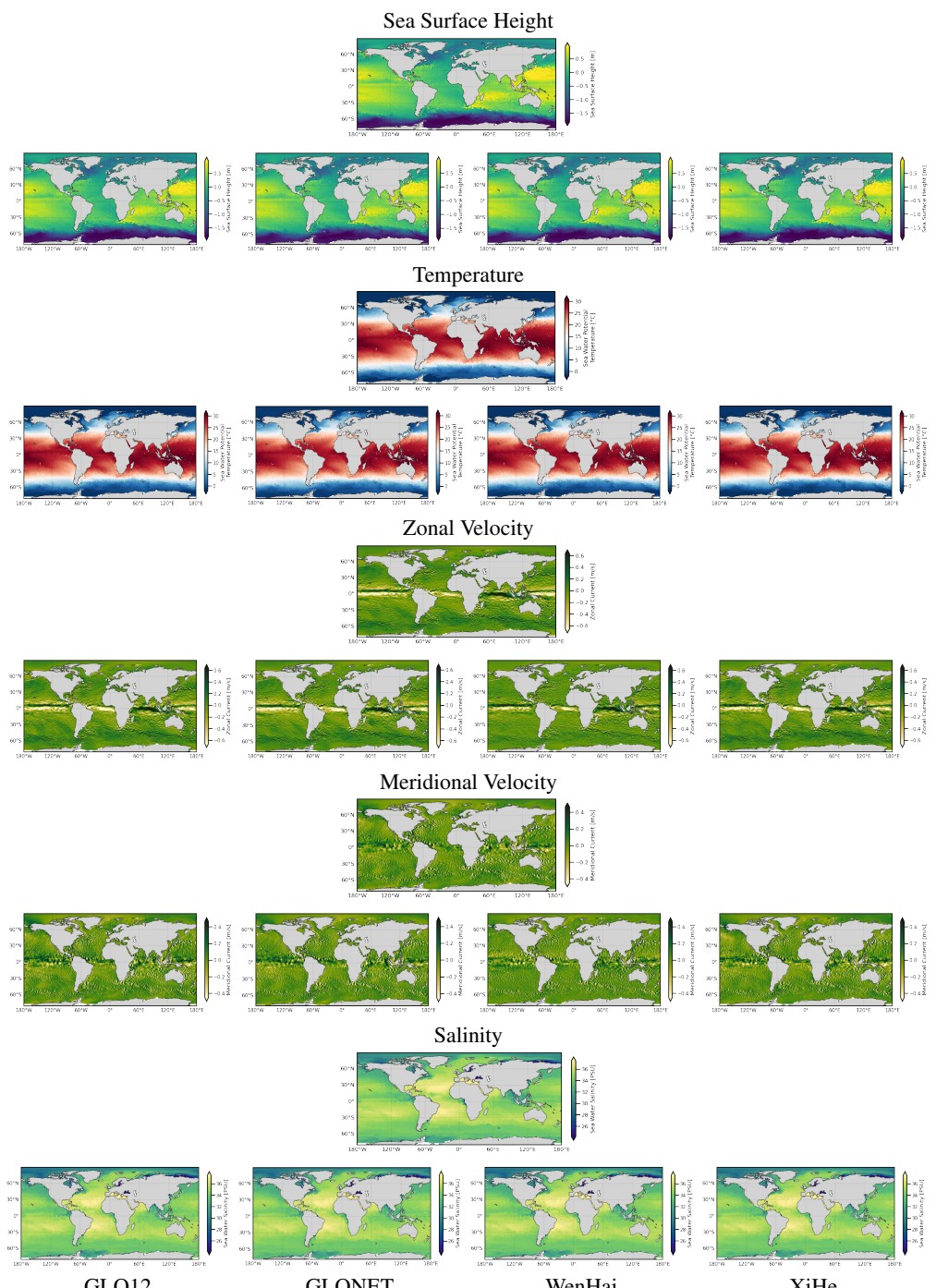

Figure 8: A set of results for world map for the forecasted variables. The following physical variables are as follows: a) Sea Surface Height, b) Seawater Potential Temperature, c) Zonal Current, d) Meridional Current, e) Salinity All of these are for lead time 1 for the date 2024-01-03.

dominant patterns of ocean variability present in the reanalysis datasets used during training or evaluation. Notably, planetary-scale wave structures are clearly visible in the surface velocity fields of some of the models, closely resembling those observed in the reference systems. These large-scale features, which are often indicative of baroclinic and barotropic wave dynamics, are generally more difficult to capture in data-driven models but appear to be well preserved across the ensemble. Their presence points to a broader capacity among models to internalize low-frequency, dynamically consistent patterns, even in the absence of explicit physical constraints or assimilation cycles. Such visual coherence provides a qualitative complement to quantitative metrics and reinforces the notion that evaluated models reproduce not only the mean state but also the spatial structure of ocean circulation.

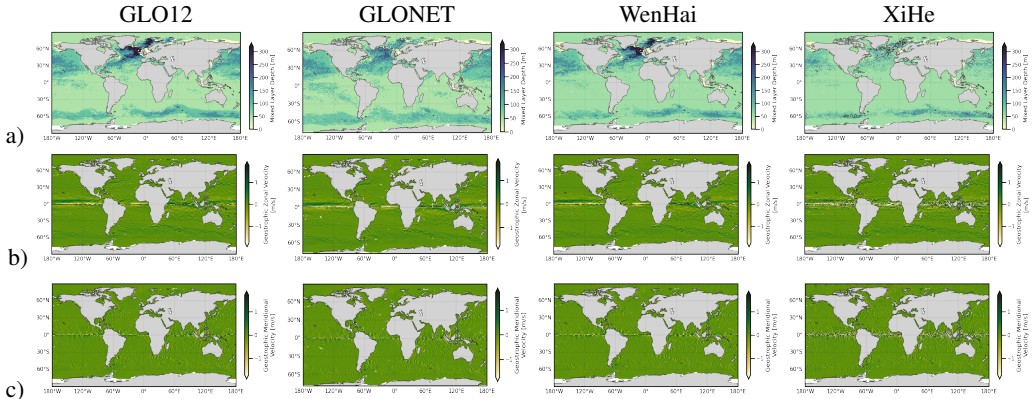

Figure 9: A set of results for world map for diagnostic variables. The following physical variables are as follows: a) Geostrophic Zonal Velocity, b) Geostrophic Meridional Velocity, and c) Mixed Layer Depth All of these are for lead time 1 on the date 2025-01-02.

When examining derived diagnostic variables such as geostrophic surface currents and mixed layer depth (MLD), the model outputs reveal varying degrees of structural coherence and persistence of artifacts (see Figure 9). In general, most models demonstrate a consistent alignment between forecasted dynamical fields and their diagnostics, suggesting a reasonable preservation of physical dependencies across variables. However, notable differences emerge based on model architecture and forecasting strategy. XiHe, employing non-autoregressive forecasting strategy, tend to exhibit reduced spatial coherence and increased noise, particularly evident in fragmented geostrophic current patterns and highly irregular MLD fields. WenHai also shows some edge-related noise in meridional geostrophic velocities near the northern and southern boundaries, but produces MLD fields that are well-structured and closely aligned with the reference GLO12. These patterns underscore the varying sensitivity of diagnostic outputs to architectural design, and highlight the value of visual diagnostics in assessing the internal physical consistency of model forecasts.

**Spatial distribution of errors.** To better understand the regional distribution of forecast skill and the origin of discrepancies, we present spatial maps of root mean square error (RMSE) relative to the GLORYS12 reference (see Figures 10-14). These maps reveal a striking degree of consistency in the geographical structure of forecast errors across models, particularly for sea surface height and salinity. Elevated errors are systematically found in western boundary current systems, equatorial regions, and zones of intense mesoscale activity, areas that are difficult to forecast due to their high dynamical variability and sensitivity to initial and boundary conditions.

While these broad spatial patterns are largely shared, noticeable inter-model differences are observed in the velocity components and temperature fields. For zonal and meridional currents, the RMSE distributions vary in both magnitude and slightly in localization, reflecting differences in how models represent and propagate dynamical features. Similarly, temperature fields exhibit some variability in error structure, likely tied to each model's handling of thermal gradients and stratification processes.

These variable-dependent differences suggest that, although models contend with common physical and observational constraints, their ability to represent ocean dynamics and thermodynamics diverges in meaningful ways. Overall, the spatial diagnostics reinforce the robustness of the benchmarking framework while highlighting the importance of evaluating model skill across individual physical variables.

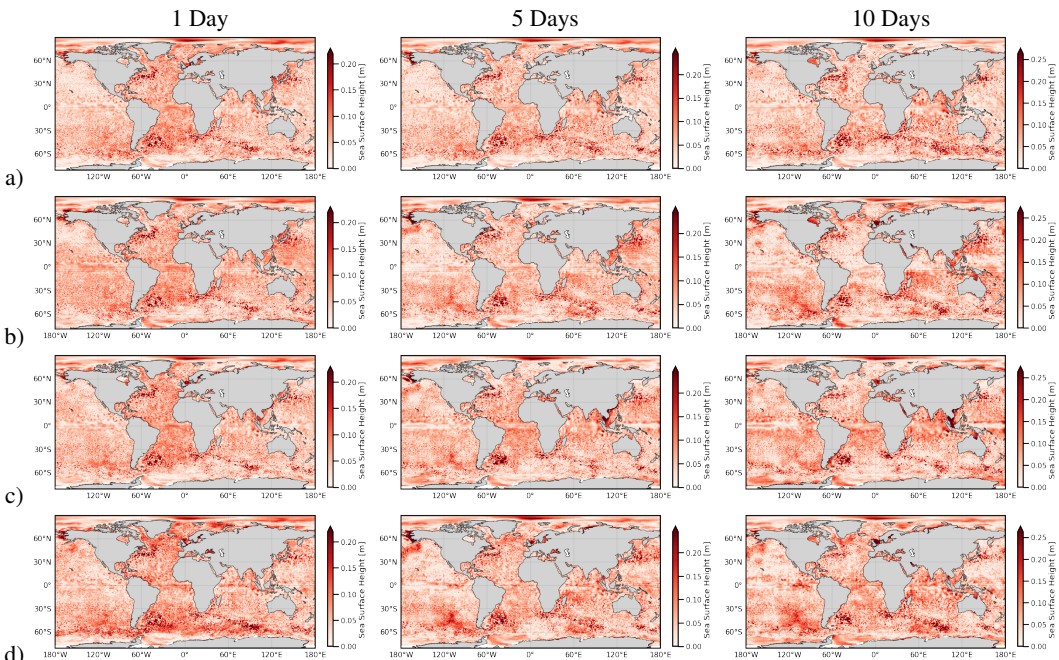

Figure 10: Error Maps of Root Mean Squared Error as a function of lead time for Sea Surface Height. Each model is compared with the GLORYS12 Reanalysis on 2024-01-03 with a lead time of 1, 5, and 10 respectively. We showcase the following variables: a) GLO12, b) GLONET, c) WenHai, d) XiHe. 2024-01-03, 2024-01-05,

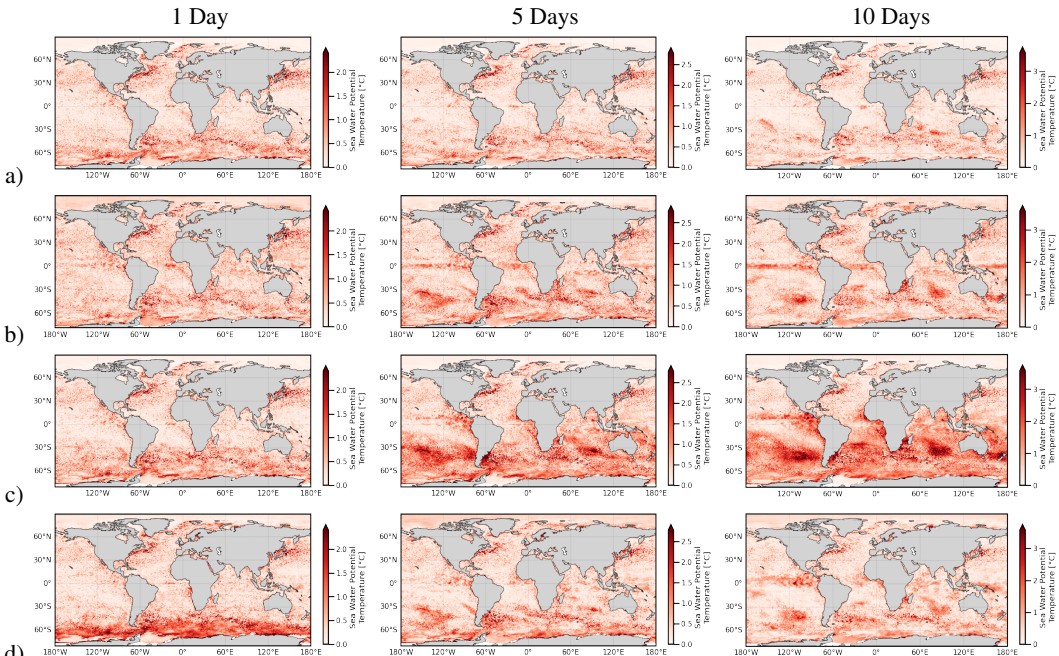

Figure 11: Error Maps of Root Mean Squared Error as a function of lead time for Temperature. Each model is compared with the GLORYS12 Reanalysis on 2024-01-03 with a lead time of 1, 5, and 10 respectively. We showcase the following variables: a) GLO12, b) GLONET, c) WenHai, d) XiHe. 2024-01-03, 2024-01-05,

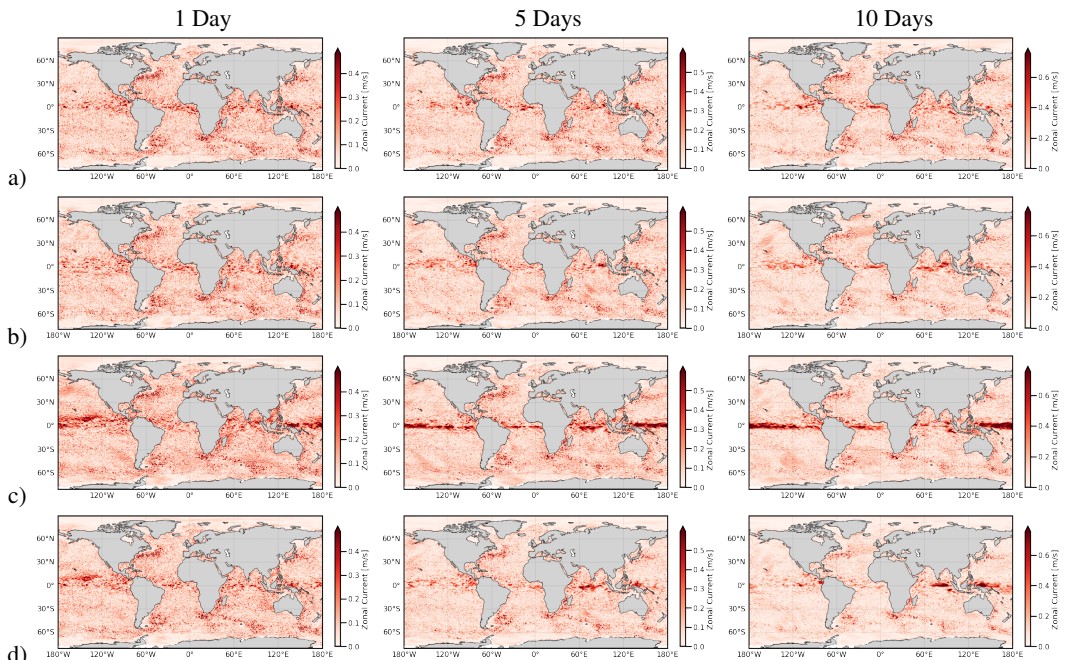

Figure 12: Error Maps of Root Mean Squared Error as a function of lead time for the Zonal Velocity. Each model is compared with the GLORYS12 Reanalysis on 2024-01-03 with a lead time of 1, 5, and 10 respectively. We showcase the following variables: a) GLO12, b) GLONET, c) WenHai, d) XiHe. 2024-01-03, 2024-01-05,

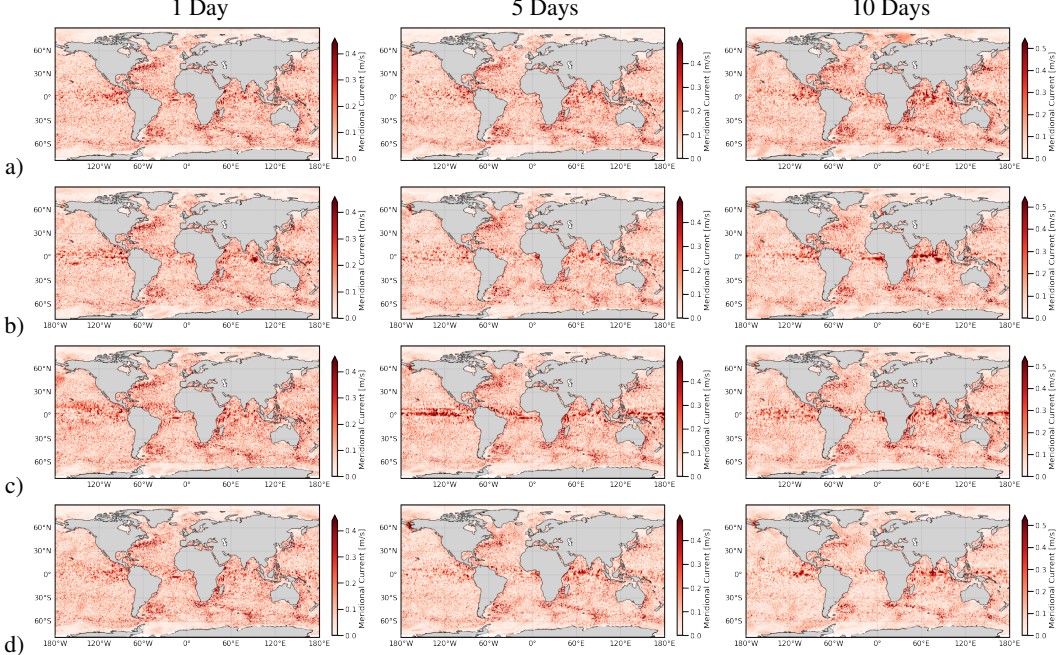

Figure 13: Error Maps of Root Mean Squared Error as a function of lead time for the Meridional Velocity. Each model is compared with the GLORYS12 Reanalysis on 2024-01-03 with a lead time of 1, 5, and 10 respectively. We showcase the following variables: a) GLO12, b) GLONET, c) WenHai, d) XiHe. 2024-01-03, 2024-01-05,

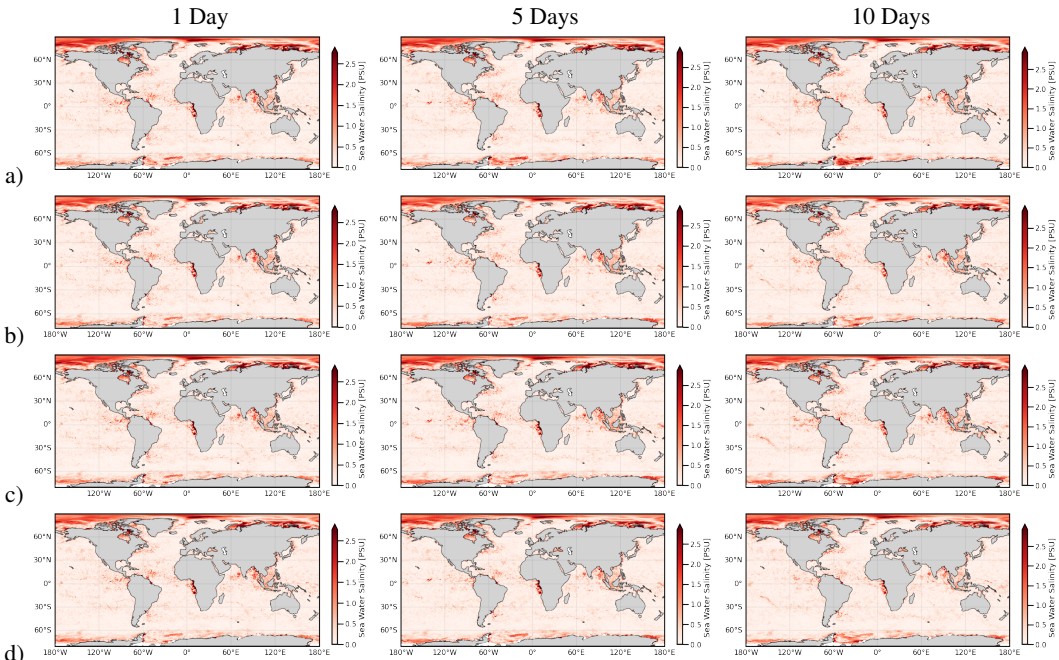

Figure 14: Error Maps of Root Mean Squared Error as a function of lead time for Sea Surface Salinity. Each model is compared with the GLORYS12 Reanalysis on 2024-01-03 with a lead time of 1, 5, and 10 respectively. We showcase the following variables: a) GLO12, b) GLONET, c) WenHai, d) XiHe. 2024-01-03, 2024-01-05,

### E.2 Power Spectral Density (PSD) Analysis

To quantitatively assess the models' ability to reproduce ocean variability across spatial scales, we analyze the power spectral density (PSD) of predicted oceanographic fields. PSD offers a robust scale-resolved diagnostic that complements RMSE and visual inspection by identifying noise artifacts, structural inconsistencies, and dynamical fidelity in model outputs.

Given a two-dimensional spatial field $f(x, y)$ defined on a regular grid, we compute its isotropic PSD as follows. First, we remove the spatial mean to eliminate the zero-frequency component: $\tilde{f}(x, y) = f(x, y) - \bar{f}$. We then apply a two-dimensional discrete Fourier transform (2D-DFT) to obtain the spectral coefficients:

$$\hat{f}(k_x, k_y) = \sum_{x=0}^{N_x-1} \sum_{y=0}^{N_y-1} \tilde{f}(x, y) \exp\left(-2\pi i \left(\frac{k_x x}{N_x} + \frac{k_y y}{N_y}\right)\right), \tag{8}$$

where $k_x$ and $k_y$ are the zonal and meridional wavenumbers, respectively. The two-dimensional PSD is given by the squared magnitude of the Fourier coefficients:

$$\text{PSD}(k_x, k_y) = |\hat{f}(k_x, k_y)|^2. \tag{9}$$

To obtain a one-dimensional isotropic spectrum, we perform radial averaging in spectral space by binning values according to the radial wavenumber $k = \sqrt{k_x^2 + k_y^2}$. This results in PSD($k$), a spectrum that describes the distribution of variance across spatial scales, enabling direct comparison of model performance in resolving fine to coarse features.

**Global-scale analysis.** At the global scale and short lead time (day 1), the PSDs reveal consistent spectral patterns that distinguish the forecasting systems (see Figure 15). WenHai stands out across most variables by exhibiting elevated spectral plateaus at high wavenumbers, indicative of pervasive high-frequency noise and a lack of effective small-scale filtering. This characteristic suggests that WenHai's outputs are generally noisier and less dynamically coherent at fine scales, even at early lead times. In contrast, XiHe shows more variable behavior: while its spectral decay in scalar fields

such as temperature and salinity is more moderate, its forecasts of vector quantities, specifically zonal and meridional velocities exhibit pronounced short-wavelength artifacts. These directional inconsistencies point to a model architecture that struggles to resolve or stabilize fine-scale dynamical structures, particularly in the representation of currents. Together, these global PSD diagnostics underscore the importance of physical constraints in mitigating high-wavenumber noise, even at the outset of the forecast horizon.

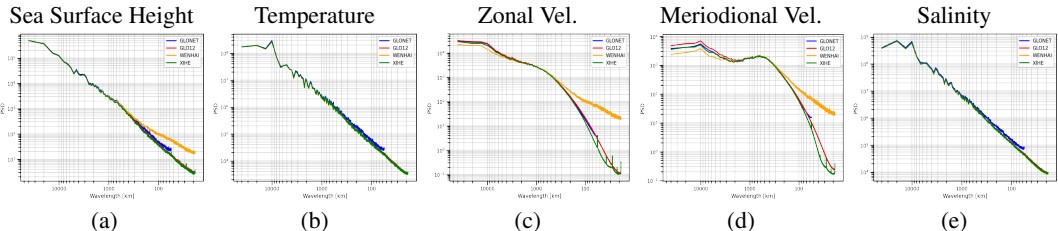

Figure 15: A set of results for the power spectrum for the zonal direction averaged over the latitude and time(2024) for the whole globe. The following physical variables are as follows: a) Sea Surface Height, b) Seawater Potential Temperature, c) Zonal Current, d) Meridional Current, e) Salinity. This figure only shows a lead time of 1.

**Regional-scale analysis.** The computed PSDs over the gulf stream region (Figure 16) reveal important differences across systems and lead times. For sea surface height (SSH), GLO12 and models architecturally aligned with it exhibit the expected monotonic spectral decay, consistent with geophysical fluid dynamics. In contrast, both XiHe and WenHai display oscillatory behavior at short wavelengths, suggestive of unresolved dynamics or spurious high-frequency noise. These discrepancies are further amplified at longer lead times (e.g., days 5 and 10), where the reduction in fine-scale energy becomes more pronounced. While such decay is a natural consequence of forecast uncertainty, the extent of energy loss and spectral distortion is particularly notable in certain ML-based systems.

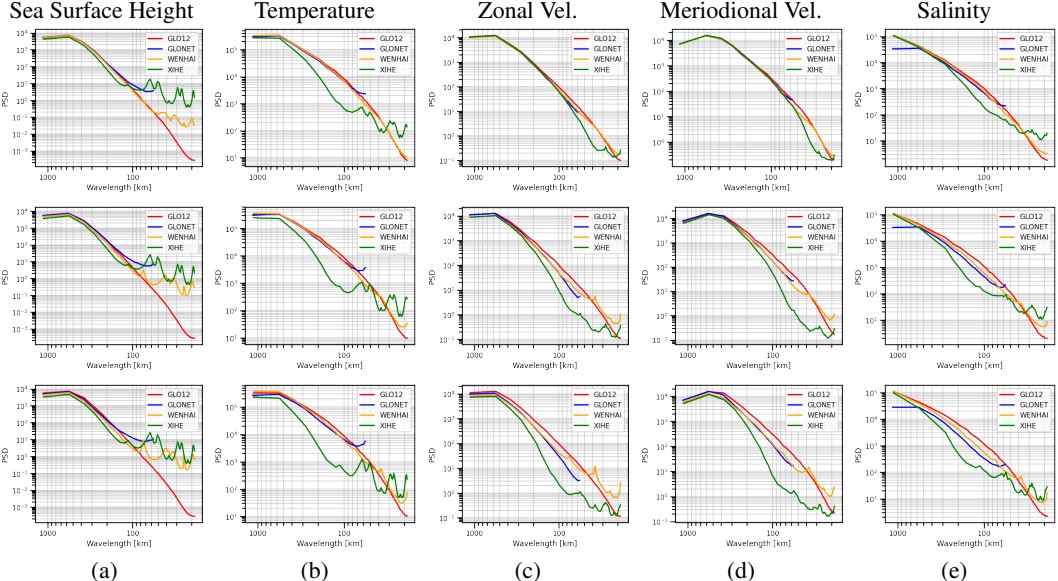

Figure 16: A set of results for the power spectrum for the zonal direction averaged over the latitude and time(2024) over the Gulf Stream. The following physical variables are as follows: a) Sea Surface Height, b) Seawater Potential Temperature, c) Zonal Current, d) Meridional Current, e) Salinity. Rows 1, 2, and 3 are the lead times of 1 day, 5 days, and 10 days respectively.

Similar spectral anomalies are observed in the temperature field, where XiHe shows both a reduced overall spectral power and pronounced short-scale oscillations, indicating a degradation in multi-scale thermal fidelity. The decline in spectral energy with lead time is consistent across systems but disproportionately affects models with weaker physical priors.

These trends extend to the zonal and meridional velocity components, where XiHe continues to exhibit the lowest spectral energy and elevated high-frequency artifacts. The salinity spectra follow a similar pattern, reinforcing the finding that some architectures are more prone to high-wavenumber noise and less capable of preserving physical structure over time.

Overall, the PSD analysis provides a rigorous and interpretable framework for evaluating the scale-resolving skill of forecasting systems. Although it does not directly disentangle specific physical processes, it effectively characterizes how each model distributes and evolves energy across spatial and temporal scales, in line with their common forecasting objective. This perspective highlights the robustness of physically grounded models in maintaining spectral coherence and exposes the challenges that certain ML-based alternatives face, particularly at extended lead times in preserving energy across scales.

