# OpenReview forum: "OceanBench: A Benchmark for Data-Driven Global Ocean Forecasting systems"
_NeurIPS.cc/2025/Datasets_and_Benchmarks_Track — NeurIPS 2025 Datasets and Benchmarks Track poster_

### Official Review · Reviewer_Ua4D · 2025-06-25

**Rating:** 5
**Confidence:** 4

**Summary:**

This manuscript introduces OceanBench, a benchmark designed for short-ranged forecast (out to ten days) of ocean dynamics, including sea surface height, temperature, salinity, and zonal/meriodional velocities in different depths. The benchmark offers various metrics and evaluation tasks against reanalysis, analysis, and observation data to assess and extend the quality of data-driven ocean models.

**Additional Feedback:**

Some suggestions and potential typos:
1. Change the name of the paper on OpenReview from all caps into the same name as used on the paper.
2. Use a single-paragraph abstract.
3. Differentiate between in-text citation and normal citation (`\citet{}` and `\cite{}` when using natbib).
4. In the last paragraph of page 2, I read _"OceanBench builds on the principles
established by benchmarks like OceanBench,"_ which appears to be an odd circular argument. Should the second "OceanBench" be "WeatherBench" instead?

**Dataset Code Accessibility:**

Yes

**Ethical Considerations:**

No, there are no or only very minor ethics concerns

**Final Justification:**

Two of my concerns were resolved: More detailed instructions how to evaluate own models on OceanBench, and a list of variables provided by this dataset.

Two other concerns were satisfactorily discussed: Latitude-weighted RMSE is crucial and will be included in future versions, and, similarly, the lead-time will be extended over 10 days in future releases.

**Limitations Weaknesses:**

1. The lead time of OceanBench is restricted to ten days, which appears short for ocean dynamics. In particular, since ocean dynamics evolve much slower than the atmosphere and the ocean is considered a major source of predictability for subseasonal to seasonal prediction of the atmosphere. It would be highly appreciated, if the benchmark would include a second task that ranges out to, e.g, six months (on a coarser temporal resolution).
2. I could not find a list of variables (and vertical levels) that are represented in OceanBench. Section 3.3 introduces the atmospheric forcing fields, but a general overview of the input variables would be convenient.
3. I was not quite sure how to evaluate my own model on OceanBench. While the [interactive notebook](https://github.com/mercator-ocean/oceanbench/blob/main/assets/glonet_sample.report.ipynb) in the repository gives some hints, I wonder what form the forecast file should have or if I'm just supposed to provide a link to my model. But how would OceanBench know how to run this model (what data resolution, step size, variables, etc. should be used)? More detailed information about how to evaluate a model would be appreciated.
4. Root mean squared error does not account for grid points of different size (assuming error calculation on the regular lat-lon grid with distortions towards the poles). Instead, a latitude-weighted RMSE would be preferable.

**Strengths Contributions:**

By tackling the prediction of ocean dynamics, the manuscript introduces an objective that is of large interest for the (data-driven) climate modeling community. The three evaluation tracks (reanalysis, analysis, observations) allow a thorough investigation of model performance beyond reanalysis data.

The repository is not only nicely structured, it is also very well documented, approachable, and accessible by offering interactive demos and easy installation instructions. Similarly, the paper is well written and organized.

OceanBench implements a reasonable model initialization procedure that does not rely on reanalysis data but on realistic model assimilation that can be realized in an operational setting. The question of initialization data has been large and controverse in the domain of atmospheric forecasts. Similar caveats seem to be prevented in OceanBench.

Scorecards in Figure 1 illustrate the qualities and limitations of different models in a very approachable and cross-comparable way.

---

> ### Author Rebuttal · Authors · 2025-07-29
>
> **Response to Reviewer's comments:**
>
>
> 1. We thank the reviewer for this insightful suggestion. The current version of OceanBench is indeed focused on short-term ocean forecasting, with a lead time of 10 days. This choice reflects both the current state of development in the ocean machine learning community and the fact that short-term forecasts (ranging from 1 to 10 days) are the standard horizon in operational oceanography, where high-resolution updates are required for applications such as maritime safety, ecosystem monitoring, and data assimilation. That said, we fully agree that subseasonal to seasonal (S2S) ocean forecasting is an important and emerging direction, both for ocean science and its relevance to long-term atmospheric predictability. As the community advances toward developing models capable of skillful longer-range forecasts, we envision that OceanBench can be naturally extended to include a second task focused on coarser-resolution, longer-lead forecasting (e.g., monthly to seasonal timescales). We have added a note in the manuscript to acknowledge this important point and signal our intention to support such tasks in future benchmark releases, in coordination with the broader community.
> 2. We thank the reviewer for pointing this out. To improve clarity and accessibility, we have added a summary table that lists all variables used in OceanBench, including their vertical levels (where applicable), units, and source datasets. This includes both the ocean variables and the atmospheric forcing fields described in Section 3.3. The new table is included in the Appendix and is intended to serve as a quick reference for users to understand the full set of inputs used across the benchmark tasks. We appreciate the reviewer’s suggestion, which helped make the dataset description more complete and user-friendly.
>
> 3. We thank the reviewer for this very helpful comment. To clarify, OceanBench supports two modes of model evaluation:
>      - Forecast Submission: users can run their models offline using the provided input data and submit their forecast outputs in a standardized format. The evaluation tools in OceanBench can then be used to score and visualize the results.
>      - Model Submission: Alternatively, users may share their model code or container along with clear instructions for how to run it (e.g., input/output format, required dependencies, etc.). In this case, the benchmark maintainers or community contributors can run the model and evaluate it using the same setup.
>
>    Some of this information is scattered between the GitHub repository, the official website, and the documentation site. To make this process more transparent, we have updated these to include 1) a detailed guide on both submission options, 2) specifications on required data resolution, time step, and variable format 3) templates and examples for forecast file structure and metadata. These additions aim to ensure consistent and reproducible evaluations while making the benchmark accessible to a broad range of users. We thank the reviewer again for raising this important point and helping us improve the user experience.
>
> 4. We thank the reviewer for this important observation. We fully agree that a latitude-weighted RMSE is more appropriate when working on regular lat-lon grids, as it accounts for the varying area represented by each grid cell, particularly near the poles. For the current version of the benchmark, we have used the standard (unweighted) RMSE for consistency with existing work and due to the time constraints associated with this release. However, we recognize the importance of improving the spatial fairness of error metrics, and we plan to integrate latitude-weighted RMSE (and potentially other area-aware metrics) in the next update of the benchmark. OceanBench is designed to evolve continuously, both by incorporating more data and by refining evaluation tools as more models and users join the community. We appreciate the reviewer’s suggestion, which is well aligned with our goals for improving the benchmark incrementally.
>
> **Response to Reviewer's  additional Feedback:**
>
> 1. Thank you for pointing this out. We have updated the paper title on OpenReview to match the formatting used in the manuscript.
>
> 2. We now use a single-paragraph abstract, as recommended.
>
> 3. We have revised citation formatting throughout the paper to consistently differentiate between in-text and parenthetical citations, following the natbib convention (\citet{} vs. \cite{}).
>
> 4. Thank you for pointing out the confusing sentence. We agree the phrasing was unclear. The intended meaning was that this version of OceanBench builds on previous domain-specific efforts (such as OceanBench: The SSH Edition) while also incorporating the completeness and robustness demonstrated in broader benchmarks like WeatherBench2. We have rephrased the sentence in the revised manuscript to avoid any ambiguity or circular reference.

---

### Official Review · Reviewer_eaa4 · 2025-07-01

**Rating:** 4
**Confidence:** 5

**Summary:**

The paper introduces OceanBench, a global ocean forecasting benchmarking tool comprising three deep learning based models. The paper also defined renalysis, operational, and observational datasets for this domain and compared the predictions generated by benchmark models to evaluate the quality in different experimental settings.

**Additional Feedback:**

1. The authors can consider a longer testing period for better evaluation.
2. The same evaluation metric is explained twice.
3. The time series RMSE figures are very difficult to interpret.
4. A reference to the IV-TT CLASS4 framework is missing. Also, check the references for other external sources.

**Dataset Code Accessibility:**

Partly

**Dataset Code Comments:**

The required library and experimental codes for all benchmark models are not available in the shared repository.

**Ethical Considerations:**

No, there are no or only very minor ethics concerns

**Final Justification:**

Many thanks for the rebuttal. I’m glad to see the authors plan to make changes to make their claims more accurate. I’ve upgraded my rating accordingly.

**Limitations Weaknesses:**

1. Need to consider more forecasting models. This paper compared three machine learning algorithms with one physics-based model.
2. The experimental design and parameters used for the comparative analysis are not clearly explained. Also, the analysis of the computational complexity and scalability is required to compare the model’s effectiveness in the spatiotemporal setting.
3. The authors only used one evaluation criterion (RMSE) for comparative analysis.
4. More evidence is required for claims articulated in the seb section 4.2. For example, “how dual evaluation strategy provides a more balanced understanding of model performance across spatial scales”.
5. There are some inconsistencies in the benchmark results explanation section, which creates confusion. Like, in the “ML versus Physics-based Models” paragraph it is mentioned that “In contrast, ML models generally exhibit lower skill on scalar tracers such as potential temperature (thetao) and salinity (so)”, which is not true for salinity.
6. Also, a precise and concrete comparison between benchmark models focusing on different dynamical processes is missing.

**Strengths Contributions:**

1. This paper explained clearly the importance and impact of the datasets and models studied here.
2. The robust experimental design demonstrates the relative analysis of physics-based simulation models with machine learning based models.
3. Consideration of multiple effective evaluation models: model with simulation, model with reanalysis, and model with real-time observations. The paper also discussed the consistency of machine learning models forecasting with the dynamic physical processes.

---

> ### Author Rebuttal · Authors · 2025-07-29
>
> **Response to Reviewer's comments:**
>
> 1. We appreciate the reviewer’s suggestion to include a broader range of forecasting models. To the best of our knowledge, the number of machine learning models for global ocean forecasting published and publicly available at the submission date was limited to the three included in the paper, reflecting the fact that this field only began to emerge in early 2024. All three relied on the same reanalysis dataset (GLORYS12) for training. Our benchmark has been built  to establish a standardized evaluation framework that enables consistent comparison across both physical and machine learning models, and as more ML and hybrid models are developed, they can be integrated into the benchmark over time, similar to the iterative expansion seen in WeatherBench. We have clarified this in the revised manuscript to better convey the intended role of this work as a foundational resource for the growing ocean forecasting community.
>
> 2.  We thank the reviewer for this helpful observation. We have revised the manuscript to clarify the experimental design and benchmark setup. In particular, we now provide a more detailed description of how each model is initialized, the forecasting setup (e.g., forecast horizon, autoregressive vs. direct prediction), and how evaluations are performed consistently in both space and time. Furthermore, we have added a brief discussion of computational considerations (Appendix), including qualitative insights on model scalability and runtime.
>
> 3.  We appreciate the reviewer’s comment. While RMSE is indeed our primary metric for comparing gridded outputs (e.g., sea surface height, temperature, salinity), we would like to clarify that additional evaluation strategies are also used, depending on the nature of the output. For Lagrangian deviation analysis, we compute the Euclidean distance between predicted and observed trajectories. For spectral analysis, we evaluate model skill using Power Spectral Density (PSD) comparisons, which assess how well different models capture energy across spatial scales. These additional metrics allow us to evaluate models beyond basic gridpoint-wise errors and are especially relevant when studying ocean processes with distinct spatial and dynamical signatures.
>
> 4. We thank the reviewer for highlighting this point. In addition to comparing raw model outputs to reference data, our benchmark evaluates derived physical proxies that reflect the internal consistency and oceanographic plausibility of the forecasts. Specifically, we compute variables such as geostrophic currents (using  sea surface height) and Mixed Layer Depth (MLD) (using vertical profiles of salinity and temperature). These are well-established oceanographic diagnostics that are sensitive to multi-variable interactions and vertical structure, and are thus valuable for assessing physical realism. Although RMSE is still used to compare these proxies to reference datasets (ensuring objectivity), the use of physically derived quantities allows us to evaluate coherence across variables and consistency with known ocean dynamics, aspects that would not be captured by naive pointwise metrics alone. As a consequence, we consider our benchmark to incorporate a dual evaluation strategy. We have revised Section 4.2 to clarify this rationale and added examples to support this interpretation. We believe this approach goes beyond traditional skill metrics and helps move the evaluation of ocean forecasting systems toward physically meaningful validation.
>
> 5. We thank the reviewer for this careful reading and valuable observation. Upon review, we agree that the original statement may be overly broad and could lead to confusion, especially given the diversity of evaluation tracks and model-reference pairings in OceanBench. To clarify: our benchmark includes multiple evaluation tracks, namely, comparisons against reanalysis, analysis, and observational datasets, and performance can vary depending on the track. In particular, while machine learning models generally show reduced skill on scalar tracers such as temperature (thetao) when compared to physical models, their performance on salinity (so) is more variable and, in some cases, competitive or even mildly superior in certain tracks. Moreover, scalar tracer performance is also strongly influenced by the underlying dynamical consistency of the forecast, which differs across model types. We have revised the text in the “ML versus Physics-based Models” paragraph to reflect this nuance more accurately and avoid overgeneralization. We now specify the context in which the statement applies and highlight that salinity results are more track- and model-dependent. We appreciate the reviewer’s attention to this detail, which helped us improve the clarity and precision of our discussion.
>
> 6. We thank the reviewer for this thoughtful observation. We would like to clarify that all benchmarked models in OceanBench are evaluated under a shared task: short-term ocean forecasting with a 10-day lead time. While the internal design of the models may differ (e.g, in architecture, autoregressive or direct, physics or ML ) they are all trained or run with the same goal in mind, and none are specialized for distinct dynamical processes. To analyze model skill across dynamical regimes, we include two diagnostic approaches that provide relevant insights:
>    - Power Spectral Density (PSD) analysis quantifies how well each model captures energy across spatial scales and lead times, which directly relates to their ability to represent key dynamical structures such as eddies, fronts, and basin-scale gradients.
>    - Time-series analysis by season offers temporal diagnostics that reveal how model performance varies across different oceanographic conditions, including seasonal dynamics of stratification, mixed-layer depth, and surface forcing.
>
>    While these tools may not directly isolate individual physical processes, they capture the 	scale-dependent and temporally varying behavior of each model in a way that is consistent with their shared forecasting objective. We have revised the manuscript to make this more explicit.
>
> **Response to Reviewer's additional Feedback:**
>
> 1. We appreciate the reviewer’s suggestion. We fully agree that a longer testing period would provide a more comprehensive evaluation, particularly to capture variability across seasons and interannual phenomena. While our current benchmark is limited to 2024 due to data availability constraints, specifically, the limited archive of forecast outputs from operational systems, we have designed the benchmark to be extensible over time. With a plan to include the full 2025 forecast cycle in a scheduled update and are working toward retrospectively adding earlier years as suitable forecast data become accessible. Expanding the temporal coverage of the benchmark is a priority for us, and we see this as essential for enabling more robust and long-term model evaluation.
> 2. Thank you for pointing this out. We revised the manuscript to remove this repetition.
>
> 3. We appreciate this feedback. We are revised the paragraph associated with the time series RMSE figures to provide better guidance for the reader and to clarify how to interpret the results.
> 4. Thank you for this observation. In the revised manuscript, we provided an appropriate citation to the IV-TT CLASS4 intercomparison framework upon its first mention in the main text, as it was previously only cited and discussed in detail in the appendix. We also reviewed the bibliography to ensure all external references are complete and properly cited.

---

> > ### Comment · Reviewer_eaa4 · 2025-08-05
> >
> > Many thanks for the rebuttal. I’m glad to see the authors plan to make changes to make their claims more accurate. I’ve upgraded my rating accordingly.

---

### Official Review · Reviewer_VpZn · 2025-07-03

**Rating:** 4
**Confidence:** 3

**Summary:**

- The paper introduces a very important benchmark framework for ocean forecasting. It is designed for global short-range (1–10 days) data-driven ocean forecasting.
- It offers curated initialization and atmospheric data, matched observational datasets for realistic evaluation, and three standardized evaluation tracks with process-oriented diagnostics. Covering key ocean variables, it enables comparisons between physics-based and machine learning models.

**Additional Feedback:**

Overall evaluation: The Paper works on a very crucial problem. However, the paper explains a lot of discussion on a high level, and lacks details.

Minor comment: The teaser figure in the paper could have been nice, which explains data format, models, types of analysis, and the full framework. Currently, it is really difficult to interpret and make the paper less comprehensible.

**Dataset Code Accessibility:**

Yes

**Dataset Code Comments:**

Authors have provided links to the code and documentation.

**Ethical Considerations:**

No, there are no or only very minor ethics concerns

**Limitations Weaknesses:**

- The paper addresses a significant problem; however, its presentation is poor. The exposition is often difficult to follow, as the authors introduce complex concepts without prior explanation, as following.
1. In Section 3.1, the paper refers to “ECMWF,” but it is unclear whether this denotes an existing model (in which case a citation is required), an evaluation metric, or something else entirely.
2. Section 4 states that all models are initialized under the same conditions with a specified frequency and forecast horizon; however, the meaning of “forecast horizon” is not defined.
3. The variables listed in Table 1 lack their possible ranges, and the term “lead time” appears without explanation.
Throughout the analysis, the authors do not indicate which figure corresponds to each discussion point, making it difficult to connect the results with the text.
4. A “Preliminaries” section, either in the main paper or an appendix, would greatly aid comprehension by providing necessary background information.
- It is challenging to distinguish between authors’ original contributions and analyses, evaluations, or metrics adopted from the literature.
- Each forecast is initialized with the same nowcast every Tuesday; however, the reason and significance of selecting Tuesday are not explained. If this follows a standard practice, a citation is needed. If the day was chosen randomly, insights from the results of other initialization days can be helpful.

**Strengths Contributions:**

- OceanBench consists of operational nowcasts/forecasts (GLO12), reanalysis products (GLORYS12), and atmospheric forcings into a single, pre-aligned package.
- It defines three evaluation tracks: Reanalysis, Analysis, and Observations, allowing assessment of both dense, full-field accuracy and sparse, resolution-agnostic observational skill.
- Provides observational data from satellite data with in-situ data, and shows strong consideration of different factors that can affect prediction.
- Have shown all types of analysis, i.e., ML models, Physics-based models, and introduced physical constraints in ML models.

---

> ### Author Rebuttal · Authors · 2025-07-29
>
> **Response to Reviewer's comments:**
>
> 1. We thank the reviewer for pointing this out. In Section 3.1, “ECMWF” refers to the European Centre for Medium-Range Weather Forecasts, an institute which operates one of the most widely used global physical atmosphere forecasting systems. In the revised manuscript, we have clarified this reference and added the appropriate citation to ensure it is clear that ECMWF refers to an institutional forecast provider and not a model name or metric. We have also reviewed the manuscript for similar instances where acronyms or technical terms could benefit from clearer definitions upon first mention.
>
> 2. We thank the reviewer for highlighting this ambiguity. In Section 4, “forecast horizon” refers to the length of time into the future that each model predicts from a given initialization. For example, a 10-day forecast horizon means that the model produces predictions for days 1 through 10 after initialization. We have clarified this definition in the revised manuscript to ensure the terminology is accessible and unambiguous to all readers.
>
> 3. We thank the reviewer for these helpful observations. In response:
>    - we have updated Table 1 to include typical value ranges for each variable, providing readers with better physical context and interpretability.
>    - the term “lead time” has now been clearly defined upon first use. Specifically, it refers to the time elapsed between the model initialization and the target forecast time (e.g., a lead time of 5 days corresponds to a prediction made at 5 days after initialization).
>    - To improve readability, we have revised the results section to explicitly reference each corresponding figure at the relevant discussion points. This should make it easier for readers to follow the narrative and interpret the results in connection with the visualizations.
>
> 4.  We thank the reviewer for this excellent suggestion. To improve accessibility and ensure clarity for a broader audience, we have added a “Preliminaries” section in the appendix, which provides key background information on ocean forecasting concepts, datasets, and terminology used throughout the paper. We believe this addition will help readers, especially those from outside the oceanography community, better navigate the technical content of the main paper.
>
> 5. We thank the reviewer for the comment. The main original contribution of this work lies in setting up a standardized benchmark for the ocean forecasting community, providing a common ground for training, evaluating, and comparing models across diverse approaches. Most of the metrics used are well-established and widely applied across geophysical and machine learning domains. Some are more specific to oceanography (e.g., metrics for mixed layer depth or geostrophic currents), and in those cases, we have gathered and adapted them to fit the structure and goals of the benchmark. Our contribution is thus in the careful selection, adaptation, and integration of these metrics into a coherent evaluation framework tailored for ocean forecasting. In addition, we have curated and standardized datasets for evaluation, including several that are not commonly available to the public or not typically used for systematic model comparison. Making these datasets accessible and reproducible within a unified benchmark pipeline is a key part of our contribution, and we believe it will help accelerate progress and reproducibility in ocean forecasting research. We have revised the manuscript to clarify this point.
>
> 6. We appreciate the reviewer’s attention to this detail. The choice of Tuesday for forecast initialization is not arbitrary. It aligns with the weekly operational cycle used by major ocean forecasting centers, where data assimilation is typically performed to produce a nowcast, i.e., the best estimate of the ocean state based on recent observations. By initializing forecasts on Tuesday, we ensure that all models start from this most accurate and physically consistent ocean state, making the comparison across systems more meaningful and fair. This practice is also consistent with physical operational models. We have clarified this point in the revised manuscript and included a reference to operational ocean forecasting protocols where appropriate.
>
> **Response to Reviewer's additional Feedback:**
>
> - We thank the reviewer for the valuable feedback. In response, we revised the manuscript to ensure it communicates clearly to a broader audience, including those less familiar with ocean forecasting. Specifically, we added new diagrams and summary tables to better synthesize key components of the benchmark, such as data, model types, evaluation metrics, and overall framework. These additions aim to complement the high-level discussion with more concrete visual support, making the structure and purpose of the benchmark easier to follow and interpret.

---

> > ### Comment · Reviewer_VpZn · 2025-08-06
> >
> > The reviewer thanks the authors for their rebuttal. I will keep my positive score.

---

### Official Review · Reviewer_5U3Y · 2025-07-03

**Rating:** 5
**Confidence:** 4

**Summary:**

This work introduces a deterministic benchmark for short term global ocean forecasting (1-10 days). For this, data and evaluation protocols are provided and used to directly benchmark 3 state-of-the-art ML models against the GLO12 global ocean forecasting system. The benchmark is carried out across three sets of targets: reanalysis data (which has been, albeit from a previous time period, been used to train the ML models), analysis data (from the GLO12 model) and direct observations of key ocean variables. The current ML models are able to outperform GLO12 in determinstic evaluation for a few variables, but still lack behind the physical model in many others.

**Dataset Code Accessibility:**

Yes

**Ethical Considerations:**

No, there are no or only very minor ethics concerns

**Final Justification:**

This work introduces a novel and timely benchmark for ocean forecasting at global scale. The paper is well written, considers many relevant aspects and is expected to become an important resource for the community. Unfortunately probabilistic forecast evaluation is only promised by the authors as a future extension of this "living" benchmark. I thus raise my recommendation and recommend acceptance at NeurIPS, with the hope that the authors keep their word and include probabilistic forecasts in the future.

**Limitations Weaknesses:**

Major comments:
1. No evaluation for probabilistic models. The authors claim this as future work, but I do not think one can really shy away. While it seems true that the ocean lacks behind the atmosphere in terms of models, it is unclear to me why one would not directly go for CRPS and other proper scoring rules. In the end, the aim of forecasting is and has always been predicting a distribution / ensemble.
2. Do you provide lower resolution versions of the benchmark? As was the case with WeatherBench? This could in particular make research easier for smaller academic labs with limited compute, and thus allow for more diversity in approaches. But also if compute is not constraint, typically lower resolution experiments are used for quick experiments during model development before final model runs.
3. I am surprised, it seems you compare against relatively few (if any) satellite observations? Why is that?
4. I miss a concrete description / table of the content of the datasets. What variables are available at which resolution (horizontal, vertical, temporal)? Please consider also that by submitting to NeurIPS, the work needs to be easily digestible for the broader ML community, and not just for Oceanographers.
5. I’d be interested in a more concrete description of the individual headline scores and for which applications they are most important. For example, which ones matter for the shipping or fishing industries? Which ones are interesting for surfers? etc. Perhaps you can even link it to cases where current predictive skill is too low to enable certain applications - and whre the skill would thus have to be to do that.
6. Just evaluating on one year may come with certain biases. Has 2024 been an exeptional year (you briefly say so on page 9, citing Jiang et al 2024)? How do slower modes of variability like ENSO influence this? If most models are only trained until 2019, why can’t you evaluate on 2020-2024?

Minor comments:

7. Your citations should be in brakcets most of the time and not in in-line mode.
8. The related work section seems somewhat shallow, especially when it comes to related efforts to model the ocean with ML models. I am sure there are many more works, and even if not all directly target global forecasting, I believe they still deserve to be mentioned.
9. What are “first-guess trajectories” (abstract)?
10. It would have been nice to have a version with line numbers uploaded for review.
11. Perhaps it would be better to call this work “Oceanbench2”? To have a more clear distinction to the previous work by Johnson et al (2023)?

**Strengths Contributions:**

1. Benchmark for Ocean (novel)
2. The benchmark is challenging (!) → ML models do not yet fully beat the operational model GLO12
3. Three-fold evaluation: against reanalysis & analysis to evaluate global fields. And against observations, to ideally have a model-independent forecast benchmark.
4. In addition to forecasted state variables, three diagnostic variables are computed and benchmarked against, which are related to the physical consistency of forecasts.
5. The paper and the code are well written.

---

> ### Author Rebuttal · Authors · 2025-07-29
>
> **Response to Reviewer's major comments:**
>
> 1. We acknowledge the reviewer's valuable point regarding probabilistic forecasting and the utility of proper scoring rules like CRPS. We concur that the ultimate aim in forecasting, especially within a geophysical framework, is to predict a distribution rather than a single deterministic outcome. However, our current study introduces a benchmark framework that assesses existing operational and machine learning models for ocean forecasting, which are presently predominantly deterministic. This encompasses both physics-based systems and cutting-edge machine learning models. Regarding CRPS, while it is a proper scoring rule for probabilistic forecasts, its application to deterministic forecasts (i.e., delta distributions) simplifies it to the Mean Absolute Error (MAE). Consequently, CRPS offers no additional insights beyond MAE in the context of purely deterministic models. Nevertheless, we strongly agree that future endeavors should integrate probabilistic ocean forecasting systems. As this field advances, particularly with the advent of stochastic ML-based models such as diffusion models and ensemble-based approaches, we intend to expand the benchmark (which extends beyond this paper) to incorporate proper probabilistic scoring metrics like CRPS and energy score.
>
> 2. We concur with the reviewer's suggestion that lower-resolution tracks would facilitate broader participation, particularly from research groups with limited computational resources, and accelerate model prototyping. Our current benchmark design is inherently resolution-agnostic, already incorporating models operating at diverse spatial resolutions (e.g., 1/4° and 1/12°). Furthermore, the underlying reference datasets are hosted on S3 buckets in the Analysis-Ready Cloud-Optimized Zarr format, provided at native high resolution, and can be readily interpolated on-the-fly to coarser grids to accommodate various modeling configurations. Consequently, participants retain the flexibility to select their desired resolution, and we will offer support as needs arise, contingent upon the feasible low-resolution options (e.g., 1°, 2°, or 4°) for the participants. As a result, we may introduce a separate benchmark track categorized by model resolution, reflecting the diversity of participating models. This initiative will contribute to standardizing evaluation across a wider spectrum of model capabilities and foster inclusive and scalable experimentation moving forward.
>
> 3. We thank the reviewer for this insightful question. Our evaluation approach follows the rationale of the Class-4 intercomparison framework widely used in operational ocean forecasting. These frameworks primarily rely on in situ vertical profile data (e.g., temperature, salinity) and drifting buoys (e.g SST and currents) rather than satellite observations for several reasons: In situ profiles offer full-depth information, enabling assessment of subsurface ocean structure and dynamics, which are crucial for many forecast applications and cannot be captured by surface-only satellite measurements. Satellite observations, while offering broad spatial and temporal coverage, are often more heavily processed, with derived products that may include model-based corrections, making them less suitable for fully independent validation unless carefully screened. Additionally, many satellite products (e.g., sea surface height, SST) are routinely assimilated into operational models, which risks circularity if used for evaluation. For these reasons, the Class-4 framework emphasizes unassimilated, minimally processed in situ observations as a robust and objective reference. In our benchmark, we follow this principle by using vertical profiles as the core evaluation dataset, while also incorporating satellite-derived sea level anomaly (SLA) as a key surface variable.
>
> 4. We thank the reviewer for this valuable suggestion. Indeed, to make the benchmark more accessible to the broader ML community, we have included in the revised manuscript a detailed description and a summary table of the dataset contents, covering available variables, their temporal frequency, vertical levels, and horizontal resolution. We agree this addition is essential for clarity and ease of use, especially for non-specialists.
>
> 5. We appreciate this insightful suggestion. In the revised manuscript, we now provide a more concrete description of the headline metrics, including which ocean processes they capture and their relevance to various real-world applications.
>
> 6. We thank the reviewer for raising this important point. The choice to evaluate on the year 2024 was largely driven by practical constraints related to data availability. Specifically, the baseline in our benchmark is an operational physical forecasting system. These systems typically overwrite older forecast outputs during their data assimilation cycles, meaning that only the most recent forecasts are retained. As a result, archived forecast products suitable for benchmarking were only consistently available for 2024 at the time of our study. That said, we fully agree that evaluating over multiple years is critical to account for interannual variability, including ENSO and other slower modes, and to ensure robustness. The benchmark is designed to be extensible, and we are already scheduling an update in late December 2025 to include the full 2025 forecast cycle. We also plan to gradually backfill additional (past and future) years as more retrospective forecast datasets become available.
>
> **Response to Reviewer's minor comments:**
>
>
> 1. Thank you for the suggestion. We have revised the manuscript to ensure that citations appear in bracketed format where appropriate, following standard style conventions.
>
> 2. We agree and have substantially expanded the related work section to include more relevant machine learning efforts in ocean modeling, including regional studies and alternative forecasting strategies beyond global prediction.
>
> 3. We appreciate the request for clarification. “First-guess trajectories” refer to the initial forward run of the ocean model starting from the best-estimate ocean state (i.e., the nowcast), typically initialized mid-week (e.g., Tuesday or Wednesday) in operational systems. We included a definition of “First-guess trajectories” in the revised manuscript.
>
> 4. Thank you for the reminder. We will ensure that a line-numbered version is included in future submissions to facilitate easier review.
>
> 5. We appreciate the naming suggestion. While our work is distinct in focus and scope, we have chosen to retain the current name to emphasize its domain-specific identity for ocean forecasting, while acknowledging prior work in the manuscript to avoid confusion.

---

> > ### Comment · Reviewer_5U3Y · 2025-08-04
> >
> > Dear authors, thanks for the rebuttal. I have a few open points, and am willing to raise my score if they are adequately addressed:
> > 1. Since you write that you have already updated the manuscript, can you please provide excerpts of the newly added content in the rebuttal?
> > 2. Deterministic vs. probabilistic: It is actually straight-forward to produce ensemble forecasts with deterministic models: just perturb inputs and/or train additional models from different random seeds. I must say I am still not convinced by this work not having any probabilistic evaluation.
> > 3. Resolution - perhaps you could release e.g. a 2° version of this dataset already with this paper, it seems to not be too much hassle for you?

---

> > > ### Author Response · Authors · 2025-08-05
> > >
> > > ## Manuscript revision
> > >
> > > Thank you for your thoughtful follow-up. As mentioned, we unfortunately cannot include figures or tables in this rebuttal, but we are happy to describe the newly added content.
> > >
> > > **I Dataset/Benchmark content:**
> > >
> > > In the revised manuscript, we now provide both a summary table and a schematic overview figure that clarify the structure and contents of the benchmark.
> > >  -  Variables included in the benchmark are:
> > >     - Sea surface height (SSH)
> > >     - 3D temperature
> > >     - 3D salinity
> > >     - 3D zonal and meridional ocean currents
> > >
> > > - These variables are available at daily resolution, across a range of vertical levels, and at horizontal resolutions of 1/4° to 1/12°, depending on the data source.
> > >
> > > The newly added overview figure schematizes all components of the benchmark, from input data formats and variables, to model types, evaluation pipelines, and metrics, making the full framework more accessible, especially for readers unfamiliar with ocean forecasting workflows.
> > >
> > >
> > > **II Headline metrics and their relevance:**
> > >
> > > We have expanded our explanation of the headline scores in the form of a table contextualizing the headline scores within real-world ocean applications, the table includes the following:
> > >
> > > - Sea Surface Height (SSH): crucial for understanding ocean circulation, coastal flooding, storm surge risk, and sea level rise, with direct implications for coastal planning and early warning systems.
> > >
> > >
> > > - Surface currents (zonal and meridional): central to forecasting marine debris drift, oil spill dispersion, search and rescue operations, and safe maritime navigation.
> > >
> > >
> > > - Temperature and salinity profiles: essential for characterizing water mass properties, vertical stratification, and heat content, impacting marine heatwaves, fisheries productivity, aquaculture, and biodiversity management, particularly in marine protected areas.
> > >
> > >
> > > - Mixed Layer Depth (MLD): a key indicator of nutrient mixing, influencing phytoplankton blooms, primary productivity, and downstream effects on fisheries and carbon cycling.
> > >
> > >
> > > - Lagrangian trajectories: included as a process-level diagnostic, reflecting how water parcels move over time, important for tracking pollutant pathways, plastic drift, biological larval transport, and ecosystem connectivity.
> > >
> > > These benchmark variables were chosen for their critical role in a wide range of ocean applications and their suitability for consistent, quantitative evaluation of ML models.
> > >
> > > **III Defining “First-Guess":**
> > >
> > > We have added a footnote to define first guess in the context of operational oceanography and data assimilation, which now reads:
> > >
> > > *In operational oceanography and data assimilation, the first guess (also known as the background) is the short-term forecast from a previous model run, typically used as the initial estimate of the ocean state before assimilating new observational data. It represents the model's best estimate of the current conditions based solely on past information and dynamical evolution. During assimilation, this first guess is combined with newly available observations to produce an updated analysis, or nowcast, that better reflects the true state of the ocean.*
> > >
> > >
> > > ## Deterministic vs. probabilistic
> > >
> > > While it is indeed possible to generate ensemble forecasts by perturbing inputs or training multiple models with different random seeds, implementing a consistent probabilistic evaluation framework was not feasible within the timeline of this paper. Training each model from scratch is computationally intensive, several of the models included in this benchmark require at least 1 to 2 weeks of training on dozens of large GPUs. While perturbing initial conditions is a reasonable approach for ML models, it is less straightforward to implement in the physical model we use as a baseline.
> > > That said, there are clear plans to include a probabilistic track in a future release of the benchmark. This will feature both ML-based ensembles and an in-house physical ensemble forecasting system, which is currently entering a pre-operational phase. We expect this update to be released around December or January.
> > > Although this benchmark is introduced as part of the current paper, it is designed to be a living resource. We are committed to keeping it updated and maintained, with regular improvements made available through its website and public Git repository. This release should be considered the first version of an evolving benchmark.
> > >
> > >
> > >
> > > ## Resolution
> > >
> > > A 1° version of the dataset will be made available with this version of the paper. To facilitate access and usability, it will be provided in Zarr format and hosted in an open S3 bucket, allowing efficient, cloud-native access for a wide range of users.

---

> > > > ### Comment · Area_Chair_Dc5B · 2025-08-07
> > > > **Reviewer 5U3Y please reply**
> > > >
> > > > A good discussion has been started here, but after you offering to consider further point the author's have attempted to provide some more info. Please write a closing comment to this discussion so the authors and I know where your opinion leaves off after the exchange.
> > > >
> > > > Many thanks,
> > > >
> > > > Paper 2566 AC.

---

### Comment · Area_Chair_Dc5B · 2025-08-04
**Unresponsive reviewers please reply to author's rebuttal**

Greetings reviewers who have not yet engaged with the authors for paper 2566,

While I thank you for your initial reviews, it is also mandatory that all reviewers complete the rebuttal acknowledgement, however this is not enough. You must actually read the rebuttal and think about it in detail. When the author's have asked questions, proposed additions to the paper or clarified elements you asked about, you should list these. Ideally each point can raise a meaningful discussion, but at the minimum I need to see a short assessment of whether your question/weakness remains relevant for you, for the purposes of meta review and making final acceptance decisions.

Please each write a comment back today (Aug 4) or at the very latest Aug 5, to give the authors one last day to have a final word before the close of Author-Reviewer discussions on Aug 6. Completely failing to engage in rebuttals will be flagged within the Responsible Review policy this year and can lead to penalties.

Thanks for your time and great help with NeurIPS!

Paper 2566 AC.

---

### Note · Authors · 2025-08-13

This work presents, to our knowledge, the first large-scale, systematic benchmark for global ocean forecasting with machine learning. By integrating high-quality datasets, standardized evaluation protocols, and diverse models into a unified framework, OceanBench provides an immediately actionable resource for performance assessment and methodological development. It also delivers the first direct comparison of state-of-the-art ML models with operational physical forecasting systems at a global scale, a reference point that has not previously existed in ocean science.

The potential impact parallels the transformation seen in atmospheric science with the introduction of WeatherBench (1, 2 and X), where a shared, evolving benchmark rapidly accelerated innovation, improved reproducibility, and fostered common ground between communities. OceanBench is poised to play the same role for oceanography, enabling rigorous comparisons, exposing methodological gaps, and guiding research toward societally relevant challenges. By lowering technical barriers and unifying evaluation practices, it invites participation from both ocean scientists and the broader ML community.

We are committed to making OceanBench a living benchmark. Planned updates include tracks for coarse-resolution configurations to enable broader institutional participation, integration of probabilistic modeling to capture forecast uncertainty, collaborative regional challenges, and extension to additional aspects of the ocean system. A major upcoming milestone is the biogeochemistry track, which will unify evaluation practices for biogeochemical variables and stimulate ML research in this critical area. Future releases will also strengthen interoperability with operational workflows, further bridging research and applied forecasting, and expand to longer-range and seasonal prediction horizons.

By anchoring current capabilities and defining a clear path for iterative improvement, OceanBench can help shape the next decade of AI-driven ocean modeling, accelerating scientific discovery, enhancing operational forecasting, and supporting applications from climate resilience to marine resource management. It establishes a common foundation from which innovation can grow, fostering a global community working toward a shared understanding of our changing oceans.

---

### Decision · Program_Chairs · 2025-09-18

**Decision:**

Accept (poster)

**Comment:**

OCEANBENCH is a set of real data and accompanying tools for the evaluation of ML approaches for ocean modeling. A number of comparison approaches, compatible with traditional atmospheric modeling, are provided. The data and tools allow a new and exciting scale of training and evaluation for environmental models.

The reviewers have largely agreed that the concepts and execution are both high quality. While they also raised some helpful and informative questions, the rebuttal and discussion resolved many. One outstanding point notable from 5U3Y, where the discussion has led to a promise of future support for probabilistic models - something that appears in the author's plans, and fair to be dealt with as future work, considering the already large scope of this paper.

In summary, I recommend acceptance of OCEANBENCH in the NeurIPS 25 DB track. I think that this novel resource will particularly benefit from a spotlight presentation.
,

===== FINAL UPDATE FROM DB Track PCs ====

The final decision for this paper has been taken by the program chairs after consultation with the SACs. All Senior Area Chairs have ranked papers according to the feedback from the AC during the review process. We decided to leave the original meta-review to reflect the opinion of the AC in light of the initial discussions with reviewers and SAC.